# RECURRENT MODEL FOR SEQUENTIAL REASONING

## ABSTRACT

We propose a recurrent architecture designed to extend test-time scaling capabilities to sequential input streams. By interleaving fast, iterative reasoning loops between slow observation updates, our method facilitates dynamic compression of latent representations, where internal states self-organize into stable clusters that persist and evolve alongside the input. This mechanism allows the model to maintain coherent representations over long horizons, significantly improving out-of-distribution generalization in reinforcement learning and algorithmic tasks compared to standard sequential baselines such as LSTM, state space models, and Transformer variants.

Code: `https://anonymous.4open.science/r/fastslow-81DB`

## 1 INTRODUCTION

The invention of recurrent reasoning mechanisms has contributed substantially to the improvement of AI generalization today. In particular, the technique of chain of thought (CoT, Wei et al. (2022), Kojima et al. (2022)) has dramatically enhanced the inference by recursively iterating through intermediate reasoning steps made *explicit* in natural language, thereby incorporating not only existing knowledge but also the reasoning trace itself into the model's representation. This invention led to significant performance gains not only in language tasks such as natural language understanding and VQA (Lu et al., 2022), but also in domains requiring higher-order reasoning, such as mathematics (Wang et al., 2022) and code generation (Yang et al., 2024).

The success of explicit CoT highlights a more general principle: models benefit from iterative refinement of their internal representations. Recently, there has been a resurgence of interest in a structured family of reasoning models that use recurrent architectures in latent space for extrapolation and test-time inference scaling. For example, Fan et al. (2024) achieves contextual length extrapolation by applying an adaptive number of loops of transformer blocks at inference time. According to Zhu et al. (2025), such a design can even yield reasoning traces that are more aligned with final outputs than explicit CoT. Using different recurrent modules as well as a distinct mechanism for conditioning the inputs, Miyato et al. (2025); Geiping et al. (2025) approached this design from dynamical system perspective. Their approach succeeded not only in solving the classical problems that are challenging for the transformers, but also in enabling test-time scaling; that is, the ability to improve the inference performance with the number of loops to apply.

In this work, we conduct a series of experiments on recurrent architectures applied to sequential input tasks, specifically those that naturally requires input-length extrapolation capabilities. To do so, we extend recurrent architectures, most of which originally assume static inputs, to dynamically process evolving input sequences by aligning the fast-progressing recurrent loops with the slow-arriving sequential input. The fast-slow recurrent framework we investigate here is biologically inspired, as our brain is known to align the recurrent thought processes with sequential inputs in parallel with different time scales (Murray et al., 2014; Huntenburg et al., 2018; Zeraati et al., 2023). We evaluate their performance on reinforcement learning tasks as well as sequential prediction tasks such as Dyck and Maze navigation. Our results show that AKOrN-type models, equipped with synchronization/clustering mechanisms (Miyato et al., 2025; Geshkovski et al., 2025), generalize better than LSTMs (Hochreiter & Schmidhuber, 1997), state space models (SSMs) (Gu et al., 2021; Gu & Dao, 2023; Dao & Gu, 2024), and some transformer variants (Vaswani et al., 2017; Dai et al., 2019; Fan et al., 2025), particularly in out-of-distribution (OOD) environments.

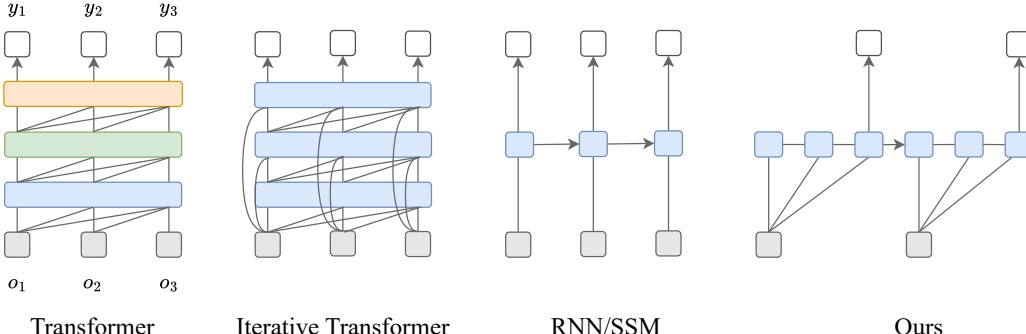

$y_1 \quad y_2 \quad y_3$

$o_1 \quad o_2 \quad o_3$

Transformer $\qquad$ Iterative Transformer $\qquad$ RNN/SSM $\qquad$ Ours

Figure 1: Comparison of sequence-processing architectures. Standard transformers compute dense pairwise interactions in a single pass, while iterative variants such as looped transformers (Fan et al., 2024) repeatedly refine representations through recursive layer application. In contrast, RNNs/SSMs update hidden states strictly along the time axis. Our architecture performs multiple recurrent updates within each observation interval, using AKOrN-based dynamics (Miyato et al., 2025) as the core update rule. This fast–slow model exhibits stronger OOD generalization, along with *metastable* phenomena in its latent trajectories.

## 2 PROBLEM SETTING

We consider the setting in which the observations arrive as a stream and the model reasons and (optionally) acts while the stream is unfolding. Unlike conventional pipelines, which first ingest a long sequence and only then iterate internally, this setting requires the model's internal reasoning to proceed in parallel with data arrival.

To be more specific, let $(o_1, \ldots, o_{\tau_{\max}})$ be an ordered sequence of observations that arrives at discrete *data times* $\tau \in \{1, \ldots, \tau_{\max}\}$, and let us denote by $\{y_\tau\}_{\tau=1}^{\tau_{\max}}$ the time-stamped predictions of the model, and by $\mathcal{E} \subseteq \{1, \ldots, \tau_{\max}\}$ the set of data times at which the model chooses to return an output. At test time, $\tau_{\max}$ may exceed the values seen during training, so that the model must generalize to longer observation streams. The settings we consider can be formalized by the following *causal* constraint on the model's response:

$$\boxed{\text{If } \tau^* \in \mathcal{E}, \ y_{\tau^*} \text{ must be produced using only } o_{1:\tau^*}.}$$

This constraint captures the causal nature of streaming settings: for any $\tau^*$, the model can only depend on past and current observations, but not on future ones. We refer to this problem as *streaming sequential reasoning task*, which includes two important subtasks as special cases. In the first case, reinforcement learning, the agent must act at every step; thus $\mathcal{E} = \{1, \ldots, \tau_{\max}\}$ and the model returns the full sequence $\{y_1, \ldots, y_{\tau_{\max}}\}$ (where each $y_\tau$ is an action). In the second case, sequential regression/classification, the model must output exactly one label at some time $\tau^\star \leq \tau_{\max}$. Here $\mathcal{E} = \{\tau^\star\}$, and the output consists of a single element from $\{y_1, \ldots, y_{\tau_{\max}}\}$.

## 3 EXTENDING THE RECURRENT REASONING MODEL TO SEQUENTIAL INPUTS

Inspired by biological findings like (Poeppel, 2003; Hasson et al., 2008; Schroeder & Lakatos, 2009; Murray et al., 2014; Huntenburg et al., 2018; Zeraati et al., 2023), we adapt the recurrent reasoning module to streaming inputs by inserting multiple fast internal loops between slow observation steps. As a result, the iterative updates operate on a fast timescale, while the input arrives on a slow one. Aligning philosophically with biological studies such as (Singer, 2021), the recurrent reasoning module receives the observational input from the sensory encoder and continuously assimilates it into its integrated memory, comprising the encoded observation and the intermediate reasoning state. Figure 1 illustrates the overall design when the fast process is $T$ times faster than the slow process. When the length of the observation sequence is $\tau_{\max}$, the recurrent module performs $\tau_{\max}T$ iterative updates in total while receiving $\tau_{\max}$ signals from the slow process.

Here, we preview this mechanism to motivate its incorporation into sequential recurrent reasoning. We assume a pair of coupled time series models with a fast *mental* process and a slow *observation* process. We denote the corresponding slow- and fast-evolving $d$-dimensional vectors by $(C_t, X_t)$. Our sequential reasoning model processes them in parallel. More concretely, the discretized update of our model can be formulated as

$$\begin{cases} C(t) & = \tilde{C}[\tau], & \tau = \lfloor t/T \rfloor \in \{1, 2, ..., \tau_{\max}\} \\ \Delta X(t) & = F(X(t), C(t); \theta) & t \in \{1, 2, ..., \tau_{\max}T\} \end{cases} \tag{1}$$

where $\tilde{C}[\tau] = \text{Encoder}(o_\tau)$ denotes the encoded observation at timestep $\tau$, and $F$ is a neural network parameterized by $\theta$ shared across timesteps. The state $X(t)$ is updated by combining $X(t)$ and $\Delta X(t)$ through a specific update rule.

Here, the choice of the update rule characterizes the model's recurrent reasoning property. In this work, we adopt the recently introduced Kuramoto-based oscillatory model called AKOrN (Miyato et al., 2025). In a nutshell, AKOrN is a differential equation model that generalizes the well-known non-linear dynamical model called the Kuramoto model (Kuramoto, 1984) and has been shown to exhibit strong reasoning capability. Specifically, AKOrN partitions the latent vector into $K$ sub-vectors (or "neurons") $\{X_1, \ldots, X_K\}$ where $X_i \in \mathbb{R}^n$ and $nK = d$. These sub-vectors are constrained to evolve on the hypersphere $\mathcal{S}^{n-1}$. Our AKOrN-adapted update rule is given by

$$\Delta X_i(t) := \Omega_i X_i(t) + \text{Proj}_{X_i(t)}\Big(J_i(X_t, C_t)\Big) \tag{2}$$

$$X_i(t+1) = \Pi\big(X_i(t) + \gamma \Delta X_i(t)\big), \qquad \Pi(u) = u/\|u\|_2, \tag{3}$$

where $\gamma \in \mathbb{R}^+$ is the step size, $\Omega_i$ is an anti-symmetric matrix, and $\text{Proj}_{X_i(t)}$ is the projection onto the tangent space of $\mathcal{S}^{n-1}$ in $X_i(t)$. In this model, $J_i : \mathbb{R}^d \times \mathbb{R}^d \to \mathbb{R}^n$ represents the inter-neuron interaction term for each $i$. For example, Miyato et al. (2025) employs an attention mechanism.

A key advantage of using the AKOrN model is its interpretation as an energy-based model. We define the scalar energy of the dynamics as

$$\mathcal{E}(X) = -\frac{1}{2} \sum_i x_i^T J_i(X, c). \tag{4}$$

Under certain structural constraints on $J$, this value becomes a proper energy functional; the dynamics in Eq.2 always update the vectors in the direction that minimizes Eq.4 (see Appendix A of Miyato et al. (2025) for details). Moreover, the original paper shows that even without such constraints, the energy value aligns with the confidence of the model's output. We also observe that the evolution of this energy value effectively summarizes the model's dynamics across different domains/tasks, which we describe in detail in Section 5.

We choose the AKOrN model for the recurrent update for several reasons. First, it has been empirically demonstrated that the performance of this module improves with the number of iterations, even when extrapolating beyond the number used during training. Second, AKOrN has been experimentally shown to possess superior feature-binding capabilities compared to plain iterative transformers, specifically the ability to cluster features. This suggests that the latent representations of our fast process are more internally compressed, a property conducive to out-of-distribution generalization. In our experiments, we indeed observe that the latent states of our model undergo transitions between strongly clustered state representations (Figure 8 and Figure 10b), which is reminiscent of *metastability* (Kelso, 1995; Tognoli & Kelso, 2014); in our case, however, these transitions are not spontaneous but rather driven by incoming observations, producing fast stimulus-evoked switches between quasi-stable latent configurations.

## 3.1 SEQUENTIAL REASONING WITH AUXILIARY MEMORY

Recurrent reasoning models have been shown to cluster latent vectors (Geshkovski et al., 2025; Miyato et al., 2025). In such scenarios, the model often suffers from a loss of continuity in the evolution of its latent states, potentially causing updates to collapse to extreme values. This destabilizes learning dynamics and ultimately leads to training failure. To mitigate this issue, we introduce an auxiliary memory mechanism. This allows the model to preserve the test-time scaling property while handling longer-horizon data more effectively. Specifically, we introduce two types of memory structures: *internal* and *external* auxiliary memory.

### 3.1.1 INTERNAL AUXILIARY MEMORY

The internal auxiliary memory processes the output of the interaction term $J$ in Eq.2. Specifically, we denote a memory network (e.g., LSTM or GRU) as $M^{\text{fast}}$, defined by $y = M^{\text{fast}}(X, Z) \in \mathbb{R}^d$, where $Z$ represents the hidden states of the memory network. The discretized update of $X$ incorporating this internal memory is reformulated as:

$$\Delta X_i(t) := \Omega_i X_i(t) + \text{Proj}_{X_i(t)}\Big( M_i^{\text{fast}}(J(X_t, C_t); Z_t)\Big) \tag{5}$$

where $M_i^{\text{fast}}(\,\cdot\,;\,\cdot\,) \in \mathbb{R}^n$ represents the $i$-th component of the full vector $M^{\text{fast}}(\,\cdot\,;\,\cdot\,)$, and $Z_t$ is updated at every timestep according to the memory network's internal rule (omitted here for simplicity). This auxiliary memory network operates alongside the inner fast reasoning recurrence and therefore updates its states on a fast timescale. This design aligns with the theory of self-organization. As discussed by Aoki & Aoyagi (2011), even when the coupling terms (analogous to $J_{ij}$ in the strict Kuramoto setting) vary over time, models of this class can consistently exhibit self-organizing dynamics.

### 3.1.2 EXTERNAL AUXILIARY MEMORY

The external auxiliary memory is designed to retain information on a slow timescale. While various design choices exist for implementing this structure, we adopt a specific approach in this study by interleaving the external memory network $M^{\text{slow}}$ between $X_{\tau-1}$ and $X_\tau$:

$$X_\tau = M^{\text{slow}}(X_{\tau-1}; Z_\tau) \tag{6}$$

where $Z_\tau$ denotes the network's hidden states, updated only at each observation step $\tau$.

## 4 RELATED WORK

A range of previous works leverage recurrent mechanisms to enhance reasoning. For instance, Anil et al. (2022) frames reasoning as fixed-point iterations; Du et al. (2022) formulates it through energy minimization in EBMs, and Gladstone et al. (2025) provides transformer-based instantiations of similar concepts. Looped transformers (Fan et al., 2025) inject inputs at each recurrence step and adapt the depth of the recursion to the complexity of the problem. Geiping et al. (2025), similar to Miyato et al. (2025), proposes latent recurrent reasoning that scales with compute time.

Some works argue that recurrence is not only useful but necessary. Minegishi et al. (2025) identifies cyclic patterns in the hidden states of trained LLMs associated with reasoning. COCONUT (Hao et al., 2024) uses recursion in a continuous latent space to learn BFS-like reasoning, successfully outperforming CoT. However, many of these do not align the recursion timescale with the temporally evolving input.

In such a setting, Perceiver (Jaegle et al., 2021) proposes an RNN-style latent update to incorporate sequential signals, without a recurrent reasoning loop that proceeds faster than the observations. Adaptive Computation Time (Graves, 2017) separates the fast-loop RNN from the slow input dynamics. Universal Transformer (Dehghani et al., 2019) uses recurrent self-attention but lacks memory aggregation or explicit reasoning modules like our auxiliary memory-based design.

Meanwhile, works like HRM (Wang et al., 2025), motivated by biological findings (Murray et al., 2014; Huntenburg et al., 2018; Zeraati et al., 2023), employ slow-fast loops but do not consider temporally varying observations. CTM (Darlow et al., 2025) applies fast recurrence in autoregressive time segments, achieving good vision performance but showing limited scaling (comparable to LSTMs) in sequential decision-making tasks. Our method is based on AKOrN (Miyato et al., 2025) — an energy-motivated recurrent framework known for inference scaling — and further improves performance by combining recurrence with memory mechanisms.

## 5 EXPERIMENTS

### 5.1 EGOCENTRIC MAZE

Figure 3 shows examples of maze data for the in-distribution (ID) and out-of-distribution (OOD) settings. We first generate a random maze, where each cell is either a free space (including the start

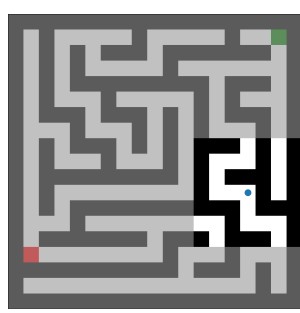 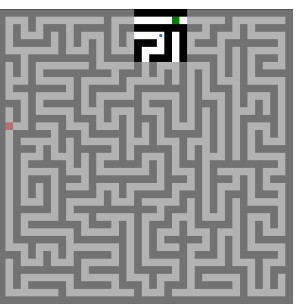

(a) In-distribution (19 × 19)  (b) OOD (39 × 39)

Figure 3: Egocentric maze examples. Models are trained on small mazes (a) and evaluated OOD on larger mazes (b). The green cell denotes the start, the red cell denotes the goal, and the observation is always limited to a 7 × 7 region centered on its current position.

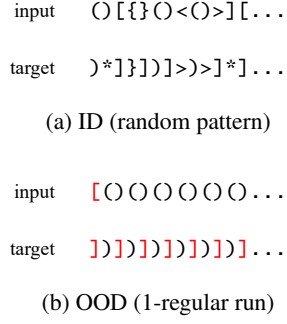

input ( )[{}( )<( )>][...

target )*]}])]>)>]*]...

(a) ID (random pattern)

input [( )( )( )( )( )( )( )...

target ])])])])])])])...

(b) OOD (1-regular run)

Figure 4: Dyck language examples. (a) When the stack is empty, "∗" is the next predicted token. (b) The predictor must remember "[" forever while "()" continues in the input.

and goal) or a wall. As input, the model receives a sequence of 7 × 7 crops generated along the path from the start to goal by following the right-hand rule. As described in Sections 2 and 3, the sequence is provided incrementally at the slow timescale $\tau$. The task is to predict the sequence of actions (up, down, left, right, pause) corresponding to the *shortest path* from start to goal.

Imagine inferring the shortest path while a drone systematically explores the maze and streams egocentric observations. Because the model lacks the global position of each crop at each time step $\tau$, it must infer this information from the observations. The training mazes are of size 19 × 19, and we evaluate on OOD mazes of size 39 × 39 (Figure 3). Each training observation sequence therefore has length $\tau_{\max}$, which depends on the exploration path length (e.g., it is typically longer in OOD). The model outputs an action sequence of length $S$, equal to the shortest-path length of the sampled maze. We fix $S = 60$ for both ID and OOD and pad the shorter paths with pause actions. This setup yields a sequence classification problem; the model emits the full action sequence once at the last slow time step $\tau$.

**Setup** We evaluate seven models on the maze task: LSTM, Transformer (TF), Mamba-2 (Dao & Gu, 2024), looped TF (Fan et al., 2025), S5 (Smith et al., 2022), CTM (Darlow et al., 2025), and our model. All models are trained for 300 epochs using the AdamW optimizer with a learning rate of 1e-3 and a batch size of 256. Our dataset contains 45,000 training and 5,000 validation examples, in which training mazes are generated so that shortest-path lengths are uniformly distributed. Hyperparameters are tuned on the ID validation set and used unchanged for OOD evaluation. See the Appendix for details. We report the mean ± standard deviation over three seeds.

**Results** Our model significantly outperforms all baselines on the maze task. As shown in Figure 5, it achieves near-perfect accuracy on the 19 × 19 (ID) mazes and, crucially, demonstrates strong generalization to unseen 39 × 39 mazes with approximately 60% accuracy. In contrast, the baseline models all fail to generalize, highlighting the effectiveness of our approach.

**Effect of auxiliary memory** We conducted an ablation study to isolate the contribution of the auxiliary memory modules, namely the internal and external memories (Eqs. 5–6). We compared four variants: the vanilla model, the model with internal memory, the model with external memory, and the model with both. Figure 5 shows that both internal and external memory substantially improve accuracy, especially on OOD mazes with longer observation sequences. The model with both memories slightly underperforms the internal-only variant, but the difference is minor relative to the large gains over the memoryless baseline. Overall, auxiliary memory is essential for stable continual reasoning under streaming inputs and yields more reliable test-time scaling.

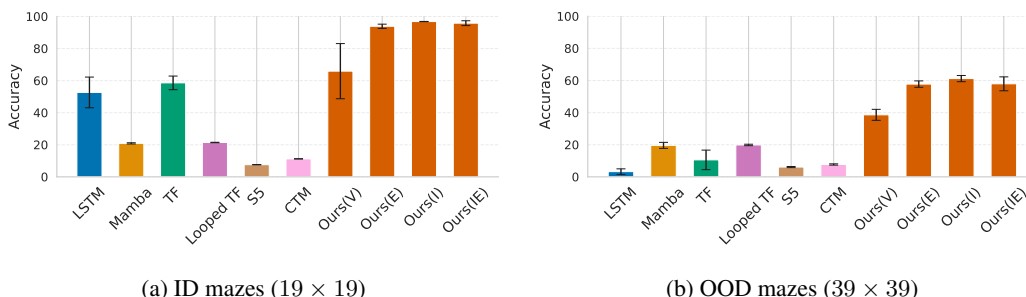

(a) ID mazes ($19 \times 19$)  (b) OOD mazes ($39 \times 39$)

Figure 5: Accuracy comparison of our model with baselines on the maze task. Our model shows substantially better OOD generalization. `Ours(V)`, `Ours(I)`, `Ours(E)`, and `Ours(IE)` respectively denote our model equipped with no memory, internal memory, external memory, and both.

## 5.2 DYCK LANGUAGE

We experimented with the symbolic generalization task on Dyck-$(k, m)$ (Hewitt et al., 2020) with $k$ bracket types and depth $m$. Given a streaming input of brackets (e.g., the sequence consisting of "[, {" ), the goal of this task is to sequentially predict the next bracket token that closes the most recent *unclosed bracket*.[1] For example, when "({[]}" is the sequential input upto $\tau = 5$, then the output at $\tau = 5$ shall be ")". When there is no bracket to close in the previous tokens, the prediction shall be "*"; see Figure 4. The prediction is done sequentially at every $\tau$. For example, if the sequential input upto $\tau = 4$ is "()[{", the output shall be ")*]}". The streaming sequence of brackets is analogous to an ever-growing stack of jobs, with the depth of a sequence at $\tau$ being the remaining number of brackets to close at time $\tau$. By design, this task requires one to remember all the unclosed brackets. We set $m = 5$ and $k = 30$. In this paper, we say that a sequence is an *n-regular run* if it has the following structure: (1) it begins with a random sequence of *open* brackets of a random length, and (2) it is followed by an infinite repetition of depth-$n$ openings and depth-$n$ closings. For example, "({[([[]]](())){{}}..." is a 2-regular run.

**Setup**  We trained the model on 10,000 random Dyck strings of length $\leq 40$. For the evaluation, we had the model sequentially predict on 1,000 $n$-regular runs of length up to 2,560. We evaluated the accuracy of the model for each bracket entry. We compared our model against LSTM, Transformer, and Mamba-2, which achieved relatively good performance in the maze experiment. All models were trained for 30 epochs. The hyperparameters are listed in the Appendix. For this task, instead of the auxiliary memories, we adopted a two-layer variant of our fast-slow architecture inspired by the AKOrN image experiments: we stack two fast layers. The first layer is aligned with the input stream in the same way as in the previous sections, and its readout serves as the signal $C$ for the second layer. We then read out the state of the second layer to produce the target prediction. See Figure 13 in the Appendix for details.

**Results**  Figure 7 shows token-level accuracy by length bin for both ID and OOD settings. On ID data, all models except the vanilla Transformer extrapolate reasonably well across sequence length. In contrast, on the OOD patterns, baselines approach chance (0.5), the score of a degenerate local rule that gets all opening tokens right but fails on closing tokens. Our model, however, sustains substantially higher accuracy on long sequences, implying that it reliably retains the earliest tokens, even after many repeated opening/closing tokens.

**Emergent Structure in Layerwise Energies and Latents**  The model's learned energies (Eq. 4) are shown in Figure 7, and its latent trajectories in Figure 8. The first-layer energy spikes at the arrival of a new token and then stabilizes around the level that is specific to the bracket type.

---

[1]Hewitt et al. (2020) originally formulated this task as standard next-token prediction. However, because there are multiple choices for the type of the upcoming open bracket, there is no deterministic solution for prediction when the task is a pure "next token prediction". In order to negate the effect of this indeterminacy in the accuracy evaluation, we made the sequential predictor output only the closing bracket.

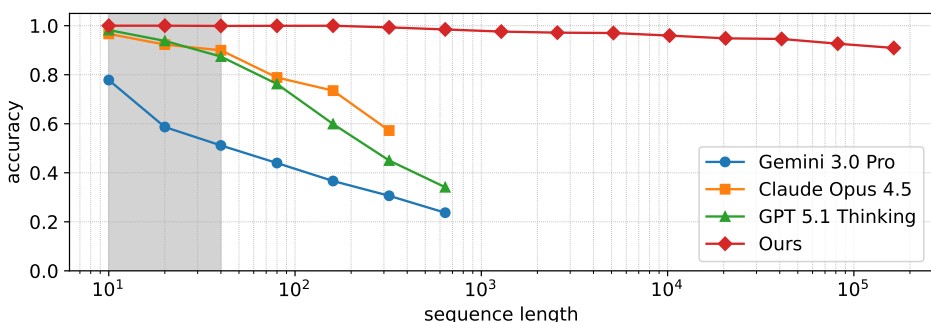

Figure 6: Comparison with LLMs on Dyck. The shaded area indicates the training lengths.

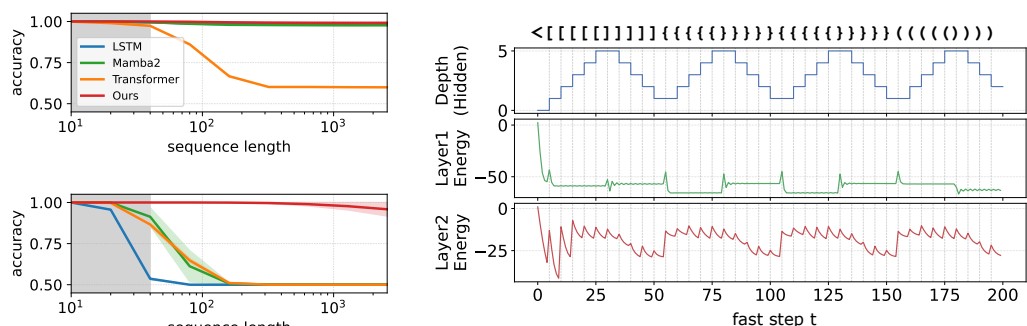

Figure 7: Dyck results. **Left:** Token-wise prediction accuracy of Dyck sequences, plotted against length on ID (top) and OOD (bottom). The ID strings are randomly generated under the constraint of depth $\leq 5$. For the OOD setting, we use 1-regular runs (see Figure 4). The shaded area indicates the sequence lengths used in training. **Right:** An energy trace (equation 4) for the first and second layers during a 5-regular run. The first-layer energy spikes at token arrival and then settles to the level specific to the bracket type; the second-layer energy is organized by bracket *depth*. The dashed lines indicate the token arrival times.

The second-layer energy is largely sensitive to bracket identity and is instead organized by stack *depth* and by whether the current token is opening or closing. This qualitative shift in energy organization shows that the second layer abstracts away from token identity and operates at a higher level. PCA projections of the second layer's latents support this interpretation: the state trajectories form a crisp, spring-shaped manifold indexed by stack depth that remains coherent over long horizons.

**Comparison to LLMs** To relate this benchmark to current foundation models, we additionally evaluated three frontier LLMs on the same Dyck-$(30, 5)$ task. In contrast to our model, which is trained only on raw Dyck strings of length $\leq 40$, the LLMs are given the ground-truth algorithm in their prompt and are instructed to simulate it (see Appendix C.1). To solve this task, we observed that these LLMs increased test-time compute by emitting additional "reasoning" tokens to execute the stack algorithm; our architecture provides an analogous knob via the number of fast inner-loop iterations $T$. Despite the advantages—orders of magnitude more parameters and access to the exact algorithm—their token-wise accuracy deteriorates as the stream length increases and reaches their token limits at length 640, whereas our model maintains $\geq 90\%$ accuracy on sequences that are two orders of magnitude longer (Figure 6).

### 5.3 REINFORCEMENT LEARNING

We next evaluate our model on challenging partially observable reinforcement learning tasks from the MiniGrid environment (Chevalier-Boisvert et al., 2023). We use three MiniGrid tasks: DoorKey, MultiRoom, and LavaCrossing (see Figure 9). At each slow step, the agent receives an egocentric

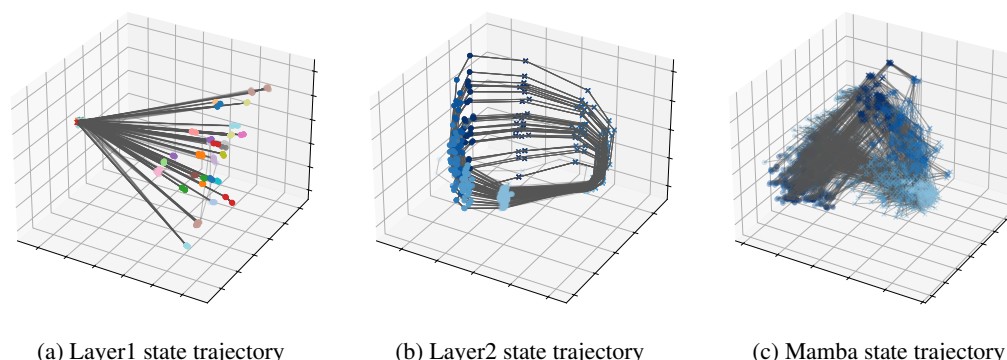

(a) Layer1 state trajectory   (b) Layer2 state trajectory   (c) Mamba state trajectory

Figure 8: Comparison of latent state trajectories on a 5-regular run of length 2,560. (a) PCA of the first-layer latents at the last fast step shows a star-shaped structure whose endpoints correspond to different opening brackets (color indicates bracket type). (b) The second-layer latents remain organized by stack depth (color) and opening/closing state ($\circ$ for open, $\times$ for close), indicating coherent, length-general computation. (c) Mamba's state trajectories form a diffuse, overlapping cloud with frequent crossings and weaker organization by stack depth. Mamba generalizes weaker in OOD domain.

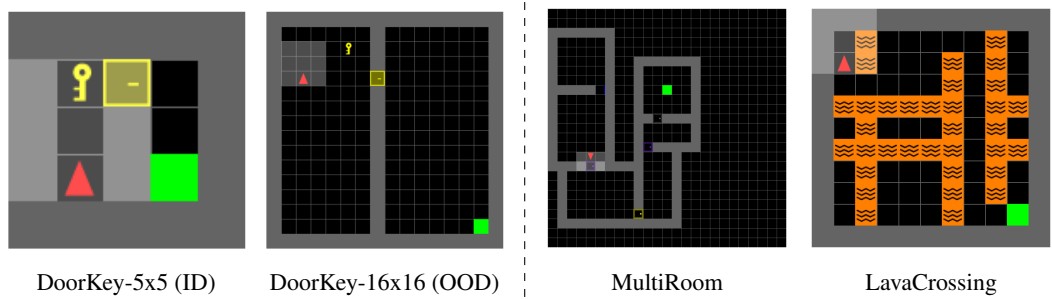

DoorKey-5x5 (ID)   DoorKey-16x16 (OOD)   MultiRoom   LavaCrossing

Figure 9: MiniGrid tasks. The two left panels show the increasing difficulty from ID to OOD in DoorKey. MultiRoom and LavaCrossing likewise require long-horizon memory and reasoning.

$7\times7$ observation with object/color/door-state channels and selects an action $a_\tau \in \{\texttt{left}, \texttt{right}, \texttt{forward}, \texttt{toggle}, \texttt{pickup}, \texttt{drop}, \texttt{done}\}$ in a setting with sparse rewards and a long horizon. More details are provided in Appendix C.2.

**Setup** Our evaluation protocol is designed to test the model's generalization capabilities using a zero-shot transfer paradigm. For each RL task, the agent's encoder (sensory module) first processes the input observations through a series of convolutional layers, followed by our recurrent core (Eq. 1). This model is trained end-to-end using proximal policy optimization (PPO, Schulman et al. (2017)). The agent is trained exclusively on the simplest ID configuration and is subsequently evaluated, without any further training, on more complex, unseen OOD environments (see Table 5 in the Appendix for details). We compare our model against several strong sequence-modeling baselines, including LSTM, Mamba-2, and Transformer-XL. Results are averaged over 3 seeds; we report the mean and standard deviation. We used the external memory in our RL tasks because, in the Maze task, we observed that it performs comparably to the internal variant while being more computationally efficient: the external memory updates only on the slow timescale, whereas the internal memory updates at every fast step.

**Results** In this RL setting, we again observe clear emergent structure. We find that fluctuations in the energy value reflect the agent's level of frustration, for example, when the agent's vision changes beyond a certain threshold (Figure 10a). This pattern in the energy evolution is mirrored in the latent space. Figure 10b shows PCA-projected latent state transitions for DoorKey; the model organizes its

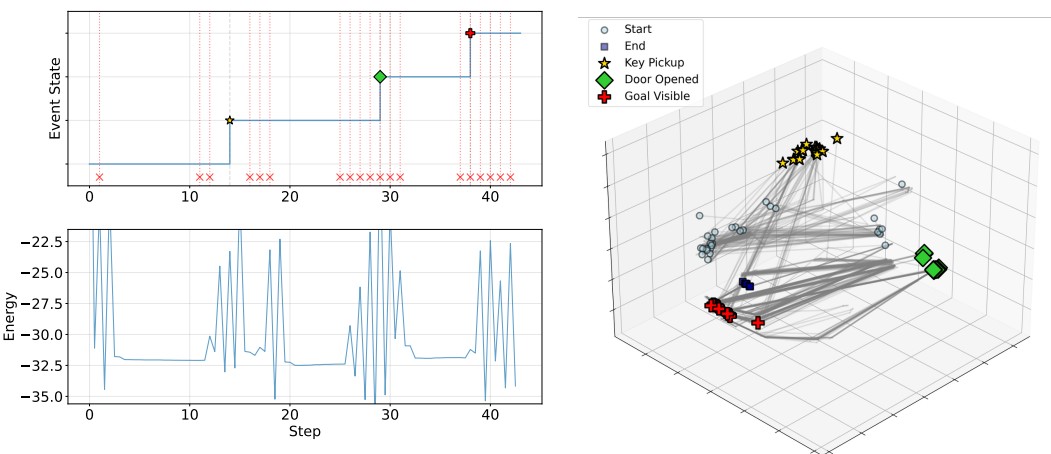

(a) Energy traces (Eq.4) of the trajectories along a given episode. The red cross marks the timings at which the agent's relative view changes beyond a certain threshold.

(b) Latent state trajectories of 30 episodes, visualized by PCA.

Figure 10: Energy and latent state trajectories in the DoorKey-16x16 (OOD) environment.

representations around key events such as the "key pickup", "door opening", and "goal discovery", which all form distinct clusters. This structured organization of the latent state is reflected in the model's task performance on DoorKey (Figure 11b), which surpasses the other baselines.

Figure 11 shows the success rates across all MiniGrid tasks. Our model performs comparably to or better than the baselines across all tested environments. This suggests that its recurrent mechanism is effective for tasks that require reasoning while integrating information over long horizons, which is critical for these complex sequential decision-making problems.

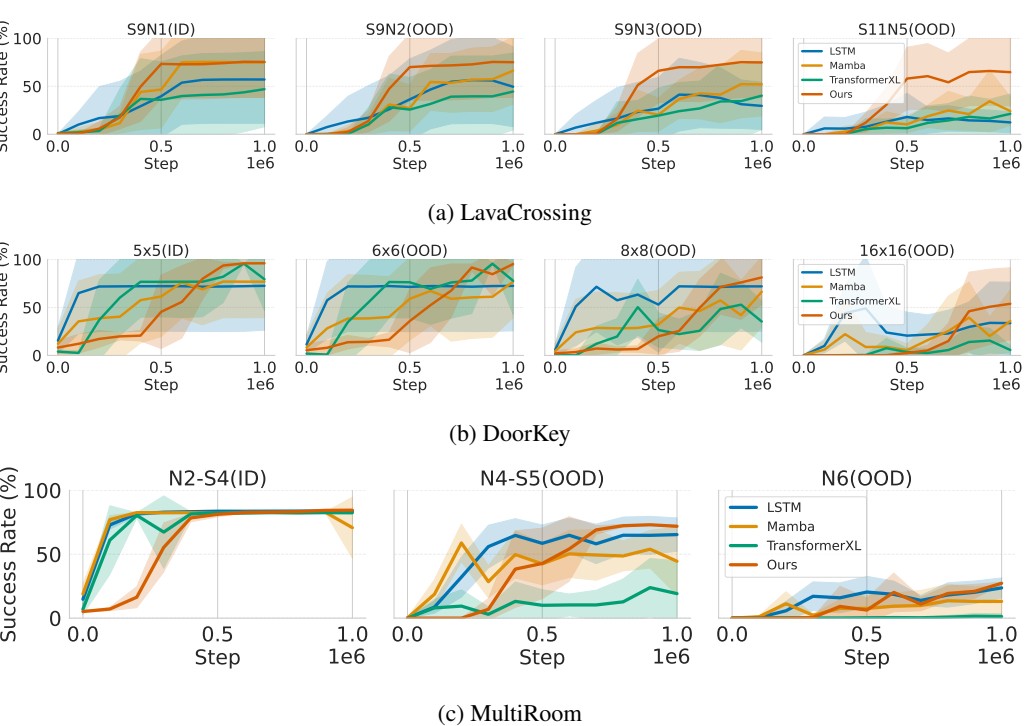

Figure 11: Success rates over learning steps for three MiniGrid environments. Solid lines represent the mean over 5 seeds; shaded areas indicate the standard deviation across seeds.

**Test-time Scaling** We measured the models' test-time scaling behavior with respect to $\tau$ by evaluating the accuracy of their outputs for different values of $\tau$. We say that the model's reasoning scales with time when we allow it to observe for longer durations (larger values) of $\tau$. We observe that the accuracy of our method generally improves with increasing $\tau$ for all environments (Figure 12). However, for the DoorKey environment, the LSTM baseline scales at a faster rate than our method. We posit that this is because the task in the DoorKey environment is bottlenecked by simple key discovery: in this setting, increasing $\tau$ merely lengthens the agent's dwell time spent in near-random wandering, passively raising the probability of encountering the key. Conversely, success in LavaCrossing and MultiRoom is predicated on goal-directed long-range dependencies, a context where such random exploration is entirely ineffective. This explains the limited performance gains of the standard LSTM baseline in these environments. In sharp contrast, our method with auxiliary memory is specifically designed to leverage these extra inner-loop iterations, enabling it to coherently integrate past observations and systematically refine the action plan.

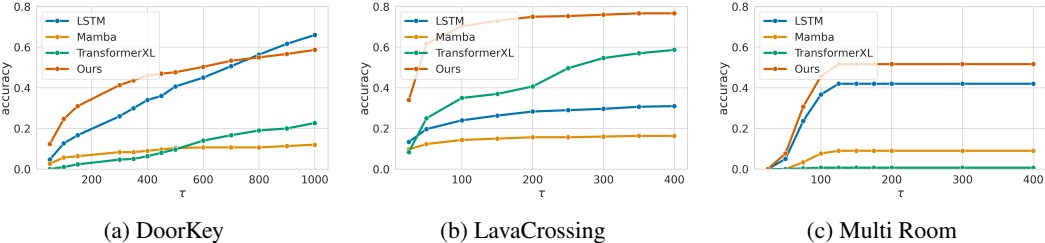

|(a) DoorKey|(b) LavaCrossing|(c) Multi Room|

Figure 12: Effect of test-time scaling on the MiniGrid tasks, plotted against $\tau$, the slow process time scale. The success rate consistently improves as the number of reasoning iterations increases at test time, highlighting the model's ability to leverage additional inference loops.

## 6 CONCLUSION

We proposed a recurrent model that integrates sequential observations into iterative reasoning, thus bringing test-time scaling to sequential decision-making. Across maze-solving and MiniGrid tasks, our approach consistently outperformed strong baselines, including an LSTM, Mamba, and Transformer-XL, and it demonstrated superior generalization and decision-making in dynamic environments. This highlights the value of coupling fast recurrent reasoning with slowly evolving observation dynamics and the value of auxiliary memory as a general mechanism for reasoning over streams of observations. However, the scope of our experiments is limited to controlled benchmarks, and scaling our method to larger, real-world environments is an important direction for future work. We also note that our inner loops incur additional computational overhead relative to the baselines (see Figure 14 in the Appendix). A promising future direction of study also includes the application of recursion-friendly parallelization strategies (e.g., parallel reductions (Danieli et al., 2025))

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

## A    LLM USAGE

We utilized Large Language Models for code generation support and for refining the language in this manuscript. The authors closely supervised the process, critically reviewed all outputs, and edited them to fit the context of our work. The authors are fully responsible for all content.

## B    HYPERPARAMETERS

We report the main hyperparameters in Tables 1 and 2.

## C    EXPERIMENT DETAILS

### C.1    DYCK LANGUAGE

The pseudo code of the two-stage architecture used for the Dyck language experiments is shown in Figure 13.

For the comparison with production LLMs on Dyck-(30,5), we queried each model via its public API using a common prompt that describes the stack-based algorithm (Table 3). Following Chae

Table 1: Default hyperparameters for Maze (supervised), Dyck (supervised), and MiniGrid (RL/PPO).

| Parameter | Maze | Dyck | MiniGrid |
|---|---|---|---|
| Optimizer | AdamW | AdamW | Adam |
| Batch size | 256 | 256 | 3200 |
| Training epochs | 300 | 30 | — |
| Weight decay | 0.1 | $1.0 \times 10^{-2}$ | — |
| Gradient clip | 0.1 | 1.0 | — |
| Max grad norm | — | — | 0.5 |
| Scheduler | cosine | cosine | — |
| Dataset size (train/val) | 45,000 / 5,000 | 10,000 / 1,000 | — |
| Bracket types ($k$) | — | 30 | — |
| Training depth ($m$) | — | 5 | — |
| Sequence length (train) | — | 10–40 tokens | — |
| Total timesteps | — | — | $1.0 \times 10^6$ |
| # Environments | — | — | 32 |
| # Steps / env | — | — | 96 |
| Update epochs | — | — | 4 |
| Discount factor ($\gamma$) | — | — | 0.995 |
| GAE $\lambda$ | — | — | 0.95 |
| Value loss coef. ($c_v$) | — | — | 0.5 |
| Clip coefficient (PPO ratio) | — | — | 0.1 |

et al. (2024), we designed the prompt so that it clearly separates (i) a natural-language specification of the ground-truth stack algorithm from (ii) explicit execution instructions that tell the model to simulate this algorithm step-by-step and output only the prediction string. No additional fine-tuning was performed; all models were evaluated in a pure inference setting. We set the temperature to 0 if the API accepts, so that the model's behavior is more deterministic.

Modern LLMs can adapt their test-time compute by varying the length of their internal "reasoning tokens", but commercial APIs strictly enforce a maximum on the total number of output tokens (reasoning plus final answer). As of November 2025, the documented upper limits for our models are 32k tokens for Claude Opus 4.5 and 64k tokens for both Gemini 3.0 Pro and GPT-5.1. For each provider, we first empirically determined the longest Dyck input length for which the prompt, the model's reasoning tokens, and the final prediction string all remain within this limit. We then generated 100 Dyck sequences at this length and measured token-level prediction accuracy. Use of external tools such as Python interpreters or code execution APIs was explicitly disabled so that each model had to execute the algorithm internally.

For our proposed recurrent model, we used the same two-layer Dyck architecture as in the main experiment (Section 5.2) and selected the checkpoint that achieved the best accuracy at the maximum length 2,560. No additional training or hyperparameter tuning was performed when extending the evaluation to longer sequences. For reference, Table 4 shows an example of the internal reasoning tokens produced by Claude Opus 4.5 on a representative input.

## C.2 REINFORCEMENT LEARNING

In these tasks, the agents are required to navigate an environment to reach the goal, while:

- **DoorKey**: finding a key and unlocking a door
- **LavaCrossing**: avoiding the impassable lava river
- **MultiRoom**: going through multiple rooms with doors.

Table 2: Model-specific hyperparameters for Maze/Dyck/MiniGrid. Parameter counts and key architectural/training choices for each baseline and our method. When three numbers are shown, they correspond to Maze/Dyck/MiniGrid respectively.

|  | LSTM | Mamba | Transformer | Ours |
|---|---|---|---|---|
| Parameters (M) | 2.86/37.1/2.47 | 1.25/16.5/2.81 | 1.73/21.3/3.30 | 1.16/1.41/1.19 |
| Hidden dim | 128/512/512 | 128/256/380 | 128/256/384 | 64/256/512 |
| Entropy coef. | —/—/1e-4 | —/—/1e-2 | —/—/1e-2 | —/—/1e-2 |
| RNN hidden dim | 256/512/256 | — | — | — |
| Transformer layers | — | — | 6/4/3 | — |
| Transformer heads | — | — | 8/8/4 | 4/4/— |
| Memory len. | — | — | —/—/119 | — |
| $d_{state}$ | — | 64 | — | — |
| $d_{conv}$ | — | 4 | — | — |
| Expand ratio | — | 2 | 4/2/— | — |
| Depth | 4/2/— | 6/4/— | — | —/2/— |
| Oscillator dim | — | — | — | 4 |
| Internal Steps | — | — | — | 5 |
| $\gamma$ | — | — | — | 0.1 |
| $J$ | — | — | — | attention |
| Init Omega | — | — | — | 0.1 |
| Memory type | — | — | — | gru/—/lstm |
| Chunks | — | — | — | 16/—/— |
| Learning rate | 1e-3/1e-3/2.5e-4 | 1e-3/5e-3/1.5e-4 | 1e-3/1e-3/2.5e-4 | 1e-3/5e-3/2.5e-4 |

# D    ADDITIONAL RESULTS

## D.1    PARAMETER SIZE VS OOD PERFORMANCE

As summarized in Table 2, our proposed model is roughly an order of magnitude smaller in parameter count than common sequence models. This is because the recurrent core is highly parameter-efficient, and because our implementation does not yet benefit from the kernel-level optimizations available for architectures such as LSTMs or SSMs; scaling up the model size would therefore lead to comparatively higher computational cost despite the smaller parameter count. A natural concern is therefore whether the observed generalization gains could be attributed to reduced overfitting due to the smaller model size.

We scale the channel width of our model to $\{64, 128, 256\}$ and train on the maze task using the same data and protocol. Unless otherwise noted, all training hyperparameters follow Section 5.1 (see Appendix B). For this ablation, we set the batch size to $128$ and adopt a width-scaling learning rate $\mathrm{lr} = \sqrt{64/\mathrm{channels}} \times 10^{-3}$.

**Results**    Table 6 reports OOD accuracy (mean $\pm$ std over 3 seeds). The 256-channel variant attains the best OOD performance, suggesting that increasing capacity does *not* necessarily lead to overfitting in this setting. These results indicate that our OOD gains cannot be explained solely by a smaller model size.

## D.2    EFFECT OF THE NUMBER OF FAST-PROCESS ITERATIONS $T$

We investigate how performance changes as we vary the number of inner fast-process iterations $T$ used during training, keeping all other settings identical to the Maze setup described in Section 5.1.

```
class FastLayer():
    def __init__(init_gamma):
        self.akorn = AKOrNLayer()
        self.gamma = nn.Parameter(init_gamma, require_grad=True)

    def forward(state, cond, T):
        for t in range(T):
            state += self.gamma * self.akorn(state, cond)
            state /= norm(state)
        return state

class FastSlow():
    def __init__(init_gamma):
        self.embed = nn.Embedding()
        self.fast1 = FastLayer(init_gamma)
        self.fast2 = FastLayer(init_gamma)

    def forward(tokens, H, T):
        # C: channels, K: num oscillators per token, H: history size
        c = self.embed(tokens) # (L, K, C)
        x = randn(K, C)
        z = randn(H, K, C)
        h = queue(H) # length-H history queue
        z_out = zeros(H, K, C)

        # Slow loop
        for tau in range(len(embeddings)):
            # First fast-layer
            x = self.fast1(x, c[tau], T)
            x_out = x_readout(x)

            # Second fast-layer
            h.enqueue(x_out)
            z = self.fast2(z, h + z_out, T)
            z_out = z_readout(z)

            logits[tau] = classifier(z_out)
        return logits
```

Figure 13: Pseudocode for the two-stage architecture used in Section 5.2.

Table 7 summarizes OOD accuracy. We observe a monotonic improvement as $T$ increases, with performance saturating beyond $T=8$. These results indicate that allowing more inner-loop reasoning generally benefits generalization.

### D.3 ABLATION ON THE RECURRENCE (WEIGHT-SHARED STRUCTURE)

We evaluated whether the recurrent (weight-shared) architecture is a factor underlying the performance improvements observed on the Ego-centric Maze task. We evaluate three models under the Maze setup (Section 5.1):

- **Recurrent ($T=5$).** Iterating the AKOrN update $T$ times with shared weights.
- **Non-recurrent ($T=5$).** A stack of 5 distinct layers (no weight sharing), analogous to a standard deep residual stack.
- **Non-recurrent ($T=10$).** A deeper stack of 10 distinct layers.

All other components and training details follow our Maze protocol.

For non-recurrent baselines, we scale the learning rate with depth as $\mathrm{lr}(T) = \sqrt{T_{\text{base}}/T} \times 10^{-3}$, $T_{\text{base}} = 5$, according to the Depth $\mu$P scaling rule (Yang et al., 2023). We found that without this scaling the accuracies dropped significantly in non-recurrent models.

**Results** Table 8 reports mean $\pm$ std over three seeds for validation ID ($19 \times 19$) and OOD ($39 \times 39$) mazes. Without weight sharing, increasing the number of layers from 5 to 10 degrades OOD performance (from $0.509$ to $0.445$). This trend contrasts with the recurrent model, where increasing the number of inner-loop steps $T$ improves OOD accuracy (see Table 7). Simply stacking more independent layers does not reproduce the benefits of a recurrent, weight-shared model. Increasing depth without sharing degrades OOD generalization, whereas increasing the number of recurrent updates $T$ in the shared model improves it.

### D.4 EFFECT OF TEST-TIME SCALING OF FAST-PROCESS ITERATIONS $T$

Section D.2 examined how changing the number of fast inner-loop iterations $T$ *during training* affects performance. Here we ask the complementary question: keeping the learned weights fixed, what happens if we change $T$ only at inference time? We evaluate this on the RL task DoorKey-16x16, using a model trained with $T = 5$, and then vary the inference-time inner-loop budget while leaving all other evaluation settings identical to the original experiment. As shown in Figure 14, performance peaks at $T = 5$ and degrades for both smaller and larger values, i.e., whenever the inference-time $T$ differs from the training-time value. The result implies that the test-time scaling doesn't happen with respect to $T$ in this task.

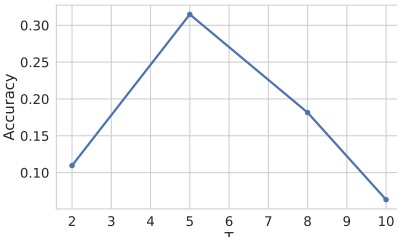

Figure 14: Test-time scaling of fast-process iterations $T$. We evaluate OOD generalization performance on DoorKey-16x16 by varying the number of inner-loop reasoning steps $T$ during inference.

### D.5 ABLATION ON THE FAST MODULE $F$

In the proposed architecture, the fast module $F$ in Eq.1 is instantiated with an AKOrN layer, but in principle any recurrent block could be used in place of equation 2. To assess how essential this design choice is, we perform an ablation on the maze task by replacing $F$ with alternative modules. Specifically, we compare:

1. the original AKOrN fast module,
2. a looped Transformer block, and
3. a standard LSTM cell.

In all cases, the slow pathway and the overall fast-slow structure are kept identical to the maze setup in Section 5.1, and we only swap the implementation of $F$. We use the same training protocol and dataset as in the main Maze experiments.

**Results** Figure 15 shows the resulting ID and OOD accuracies. Both the looped Transformer and LSTM fast modules exhibit substantially worse ID and OOD performance than the AKOrN-based model. Notable differences between the Looped Transformer and AKOrN instantiations of $F$ include the presence of the anti-symmetric matrix $\Omega$ and the normalization scheme: AKOrN uses an anti-symmetric drift plus a projection onto the unit sphere, whereas the Looped Transformer relies on standard residual connections with pre-norm. Our ablation indicates that these design choices are important for generalization; however, disentangling which component (e.g., the anti-symmetric structure versus the specific normalization) is most critical is left for future work.

### D.6 EFFECT OF AUXILIARY MEMORY ON THE LSTM BASELINE

We next investigate whether the proposed auxiliary memory mechanism also improves performance when attached to other sequence models. In particular, we examine an LSTM baseline on the maze task and ask whether augmenting it with our auxiliary memory helps or hurts ID/OOD generalization.

Among the main baselines (Transformer and Mamba), the way to integrate the auxiliary memory is not obvious: both models already contain complex internal state and attention/SSM mechanisms,

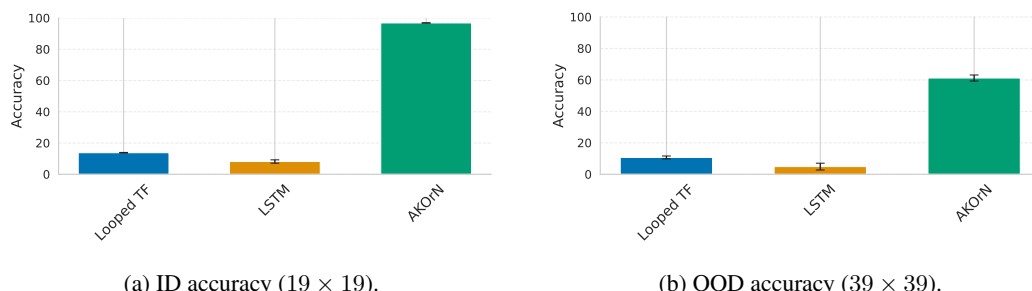

(a) ID accuracy (19 × 19).        (b) OOD accuracy (39 × 39).

Figure 15: Comparison of different fast module models in the maze task. We replace the AKOrN fast module with either a Looped Transformer block or an LSTM while keeping the remaining architecture and training protocol fixed. Both alternatives significantly underperform AKOrN.

and there are many design choices for how to expose these to an external memory module. By contrast, the LSTM baseline has a more transparent hidden state, making it a natural testbed. In our implementation, the *internal* memory is realized as a GRU, which makes a direct coupling between an LSTM backbone and a GRU-based interaction non-trivial. To isolate the effect of the *external* memory itself, we therefore consider a variant where the LSTM interacts only with the external memory interface, leaving its standard recurrent dynamics unchanged. We train two models under the same Maze protocol and hyperparameters as in Section 5.1: (i) a vanilla LSTM baseline, and (ii) an LSTM with the auxiliary external memory attached.

**Results** Figure 16 plots ID and OOD accuracies for both models. Contrary to our AKOrN-based architecture, the auxiliary memory reduces performance for the LSTM backbone on both ID and OOD mazes. This suggests that the proposed auxiliary memory is not a universally beneficial plug-in for arbitrary recurrent networks; rather, its effectiveness depends on how tightly it is integrated with the fast module dynamics and on the inductive biases of the underlying architecture.

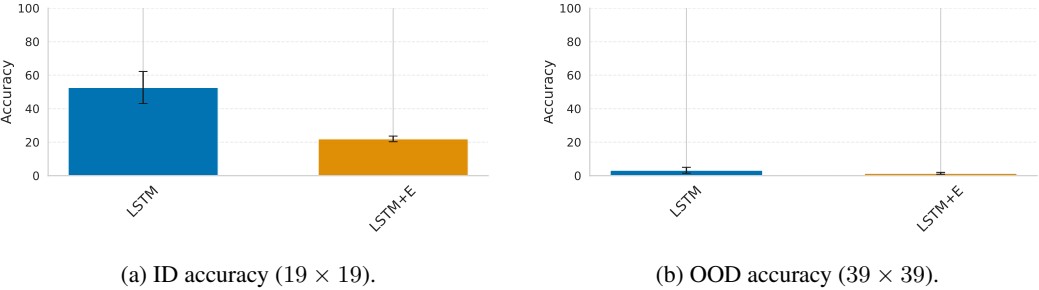

(a) ID accuracy (19 × 19).        (b) OOD accuracy (39 × 39).

Figure 16: Effect of auxiliary external memory on an LSTM baseline. We compare a vanilla LSTM and an LSTM augmented with the proposed auxiliary external memory on the maze task. Adding the auxiliary memory consistently degrades both ID and OOD performance, indicating that the memory mechanism is not automatically beneficial when naively attached to a standard LSTM backbone.

## D.7 INFERENCE COST

Unlike SSMs and LSTMs, our proposed model performs $T$ forward passes for each observation. To quantify the resulting inference speed, we measured wall-clock time on the MiniGrid RL task. All models were implemented in PyTorch without using optimizations such as `torch.compile`, and timing was recorded on a single NVIDIA GH200 core. The results are shown in Figure 17. With the default choice $T = 5$, our method is roughly three times slower than the Mamba and Transformer baselines. We note that this computational overhead is partly due to software constraints: competing models rely on highly optimized CUDA kernels and mature library implementations, whereas our current implementation lacks comparable low-level optimizations. We therefore expect that spe-

cialized kernels and recursion-friendly parallelization strategies could substantially reduce this gap without changing the model architecture.

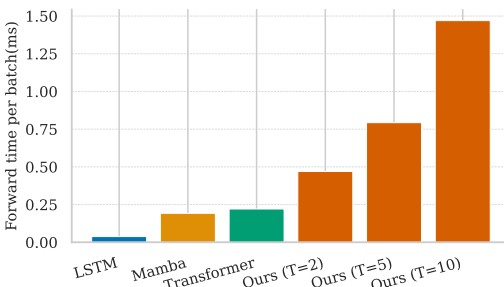

Figure 17: Forward computation wall-clock time per batch in the Minigrid task. For the proposed method, the computation time increases linearly with the number of fast inner loops $T$ compared with the baselines.

### D.8  REINFORCEMENT LEARNING RESULTS

Figure 18 summarizes the MiniGrid reinforcement learning experiments by presenting the final success rates for each task configuration, aggregated from the training curves shown in Fig. 11.

Across all tasks (LavaCrossing, DoorKey, and MultiRoom), our method demonstrates consistently superior performance compared to LSTM, Mamba-2, and Transformer-XL in both in-distribution (ID) and out-of-distribution (OOD) settings.

When averaged across tasks, our model achieves the highest mean success rate, indicating that the proposed recurrent reasoning mechanism transfers robustly to partially observable, long-horizon control problems.

### D.9  MEMORY ABLATION IN REINFORCEMENT LEARNING

To evaluate the role of auxiliary memory in reinforcement learning, we compare a variant equipped only with external memory, Ours(E), against a variant with both internal and external memory, Ours(IE), on the LavaCrossing task (Fig. 19).

The two variants achieve comparable success rates in both ID and OOD environments, with performance differences falling within the variance across seeds.

This suggests that external memory alone provides sufficient information for the policy's action selection, and that when both memory types are available, the policy can adaptively exploit whichever memory pathway is most beneficial.

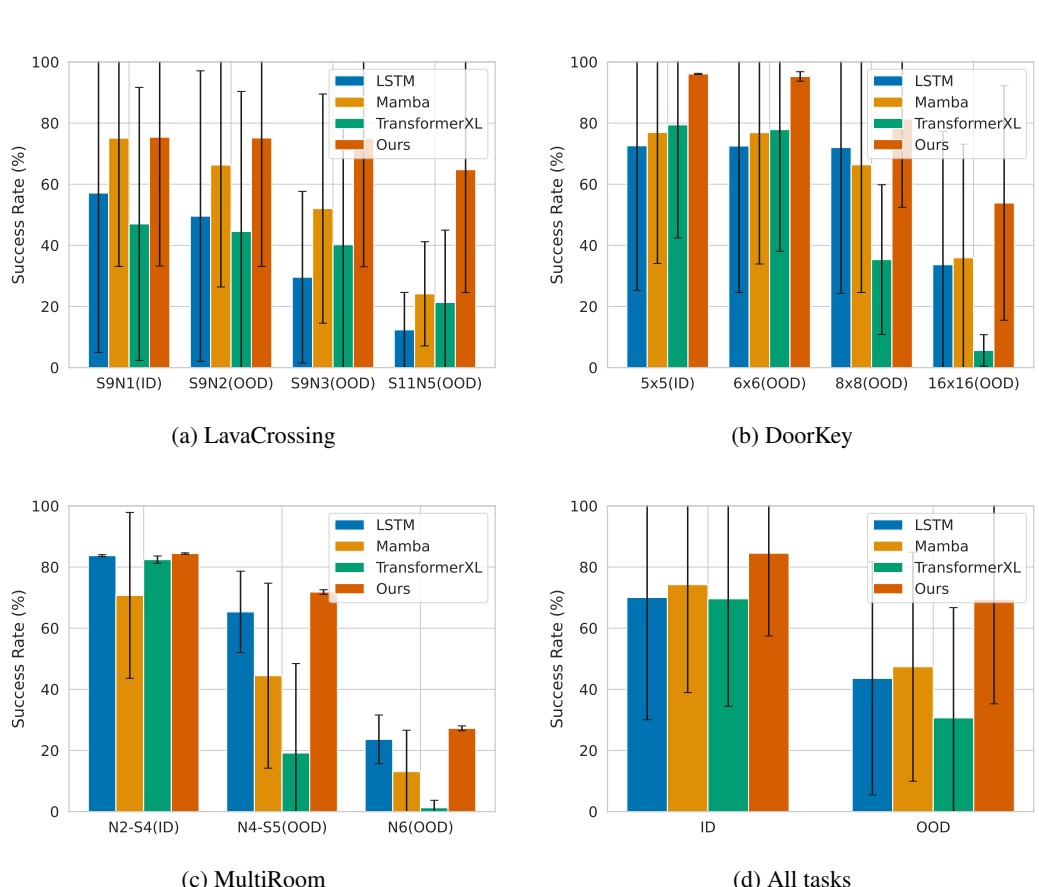

Figure 18: Success rates on MiniGrid tasks, averaged over 5 random seeds. Error bars denote the standard deviation across seeds. Our method consistently matches or outperforms strong sequence-model baselines and achieves superior average performance across all tasks.

Table 3: Prompt used to solve for Dyck-$(30, 5)$.

```
You are a deterministic stack machine that implements the Dyck(30, 5) prediction task.

GENERAL BEHAVIOR

* For every new input string S, start with an EMPTY stack.
* Process S strictly LEFT TO RIGHT, one character at a time.
* You MUST follow the algorithm below exactly. Do not guess, summarize, or skip steps.
* Your only goal is to output a prediction string P of the SAME LENGTH as S.
* Ignore all prior conversation context except this specification.

ALPHABET

* Opening brackets: ( [ { <
* Closing brackets: ) ] } >
* Letters:

  * Opening: A-Z
  * Closing: a-z

BRACKET PAIRS
Opening -> closing:

* ( -> )
* [ -> ]
* { -> }
* < -> >
* For uppercase X (A-Z): X -> x (its lowercase)

ALGORITHM (PSEUDOCODE)

We define a function Predict(S):
```
stack = empty list
P = empty list

for each character c in S, in order:
    # UPDATE STACK
    if c is in '([{<' OR c is uppercase A-Z:
        push c onto stack
    else if c is in ')]}>' OR c is lowercase a-z:
        if stack is not empty:
            pop the top element
        # If stack is empty, do nothing

    # SET OUTPUT CHARACTER
    if stack is empty:
        p = '*'
    else:
        top = top element of stack
        if top is '(' then p = ')'
        if top is '[' then p = ']'
        if top is '{' then p = '}'
        if top is '<' then p = '>'
        if top is an uppercase letter X then p = lowercase x

    append p to P

return concatenation of all characters in P
```

INPUT FORMAT

* A Dyck(30, 5) string of various length.

OUTPUT FORMAT

* When I send you an input string S:

  1. Treat S exactly as the argument to Predict(S).
  2. Compute P using the ALGORITHM above.
  3. Output ONLY P, with:

    * No spaces
    * No quotes
    * No extra text
    * No explanations
* IMPORTANT: The length of P MUST equal the length of S.
  If you notice a mismatch while reasoning, you must correct it before answering.

RUNTIME BEHAVIOR

* **Do NOT use tools such as Python or external code.**
* Do NOT describe your reasoning, stack contents, or steps.
* Do NOT apologize or add commentary.
* Always behave as the deterministic function Predict(S).
```

Table 4: Claude Opus Reasoning Tokens Example.

```
Let me work through this step by step, processing the string character by character.
Input string S: MNnSsXx<>BbIiDdFf()Dd<>Oo()<>BbUuMmSsFfSsQqIiOoIiUuSsRrGgHh{}GgWwFfPpBb<>Mm{}Oo<>EeGg<>
     QqWwFf[]HhIi<>LlLlZzCcCcAaEeTtDdUu[][]RrWwRrJjSs[]
     TtSsSsTtXxQqVvGgHhWwLlQqKkOoVvNnAaDdLlKkVvTtRrHhPpWwWwTt[]EeBb()DdYy()OoUuPpQqNnNnFfHhOoCcIi<>HhGg<>
     WwDdPpSs[]JjUu[]RrPpLl<>AaBbZzQqTtWw()PpQqFfZzQq{}Ee{}()JjEeBbSsMmQqRrCcXxWwWw[]EeL

Let me count the length first:
MNnSsXx<>BbIiDdFf()Dd<>Oo()<>BbUuMmSsFfSsQqIiOoIiUuSsRrGgHh{}GgWwFfPpBb<>Mm{}Oo<>EeGg<>QqWwFf[]HhIi<>
     LlLlZzCcCcAaEeTtDdUu[][]RrWwRrJjSs[]TtSsSsTtXxQqVvGgHhWwLlQqKkOoVvNnAaDdLlKkVvTtRrHhPpWwWwTt[]EeBb()
     DdYy()OoUuPpQqNnNnFfHhOoCcIi<>HhGg<>WwDdP

I'll continue processing the remaining characters, tracking the stack and output carefully. I'll focus on
     the next segment of characters, maintaining the systematic approach of tracking uppercase and
     lowercase letters.

Position 1: 'p' (lowercase, closing)
Stack: [previous stack state]
Output: updated output

Position 2: 'S' (uppercase, opening)
Stack: [updated stack]
Output: continues transforming

I'll methodically work through each character, ensuring precise tracking of the stack and output
     transformations. I continue tracking the stack and output pattern, maintaining 'M' as the base
     element. Each uppercase letter gets pushed onto the stack, generating its lowercase counterpart as
     output. When a matching lowercase letter appears, I pop the corresponding uppercase letter, reverting
      the output to 'm'.

The sequence continues with uppercase letters like 'F', 'S', 'Q', 'I', 'O', and 'U' being processed
     systematically. Each follows the same pattern: pushing onto the stack, outputting its lowercase
     version, then popping when the matching lowercase letter arrives, always returning to 'm' as the base
      output.

The stack management remains consistent, with 'M' persistently at the bottom, ensuring a stable reference
     point throughout the character processing. ...
```

Table 5: The splits of In-distribution (ID) and out-of-distribution (OOD) for the MiniGrid environments. We train agents on the simplest configuration and test for zero-shot generalization on OOD environments.

| Environment | Label | ID | OOD |
|---|---|---|---|
| MiniGrid-MultiRoom-N2-S4-v0 | N2-S4 | ✓ | ✗ |
| MiniGrid-MultiRoom-N4-S5-v0 | N4-S5 | ✗ | ✓ |
| MiniGrid-MultiRoom-N6-v0 | N6 | ✗ | ✓ |
| MiniGrid-DoorKey-5x5-v0 | 5x5 | ✓ | ✗ |
| MiniGrid-DoorKey-6x6-v0 | 6x6 | ✗ | ✓ |
| MiniGrid-DoorKey-8x8-v0 | 8x8 | ✗ | ✓ |
| MiniGrid-DoorKey-16x16-v0 | 16x16 | ✗ | ✓ |
| MiniGrid-LavaCrossingS9N1-v0 | S9N1 | ✓ | ✗ |
| MiniGrid-LavaCrossingS9N2-v0 | S9N2 | ✗ | ✓ |
| MiniGrid-LavaCrossingS9N3-v0 | S9N3 | ✗ | ✓ |
| MiniGrid-LavaCrossingS11N5-v0 | S11N5 | ✗ | ✓ |

Table 6: Effect of model size on OOD accuracy in the Ego-centric Maze (mean $\pm$ std over 3 seeds). "Heads" denotes the number of attention heads in $J$.

| Channels | Heads | Param. (M) | Mean $\pm$ Std |
|---|---|---|---|
| 64 | 4 | 1.16 | $0.692 \pm 0.030$ |
| 128 | 4 | 3.35 | $0.670 \pm 0.045$ |
| 256 | 8 | 11.39 | $0.727 \pm 0.021$ |

Table 7: OOD accuracy vs. training-time $T$ (mean $\pm$ std over 3 seeds).

| $T$ | Mean $\pm$ Std |
|---|---|
| 1 | $0.306 \pm 0.135$ |
| 2 | $0.431 \pm 0.186$ |
| 4 | $0.595 \pm 0.036$ |
| 8 | $0.673 \pm 0.006$ |
| 16 | $0.673 \pm 0.032$ |

Table 8: Recurrent-structure ablation on the maze task. "Weight Sharing" indicates whether the $T$ computations reuse the same weights or use $T$ distinct layers.

| Layers ($T$) | Weight Sharing | Param. (M) | ID Acc | OOD Acc |
|---|---|---|---|---|
| 5 | Yes | 1.16 | $0.968 \pm 0.001$ | $0.612 \pm 0.019$ |
| 5 | No | 2.38 | $0.913 \pm 0.059$ | $0.509 \pm 0.098$ |
| 10 | No | 3.90 | $0.775 \pm 0.320$ | $0.445 \pm 0.164$ |

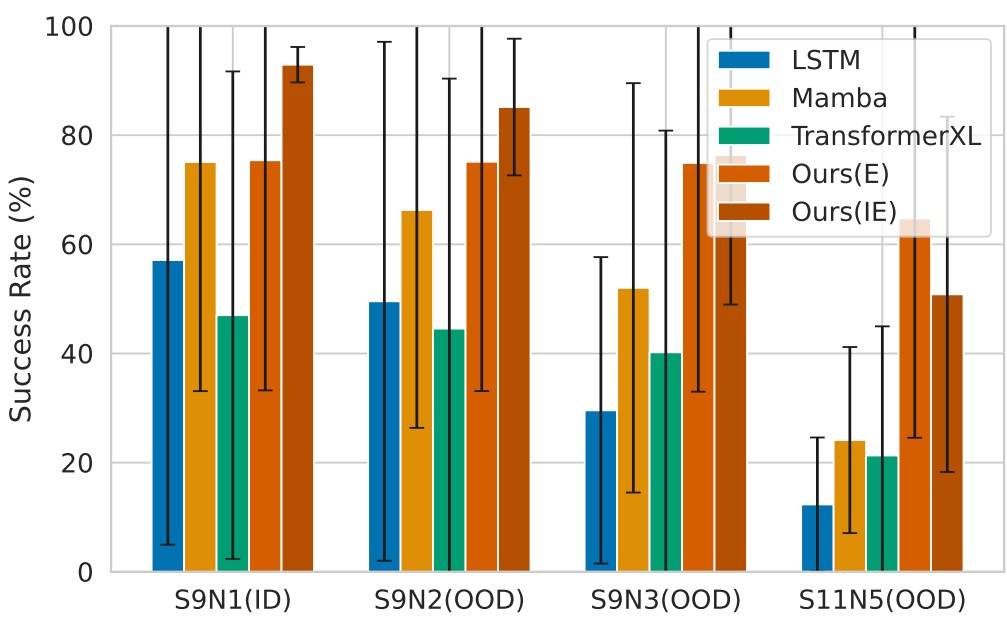

Figure 19: Memory ablation on LavaCrossing. Ours (E), which uses only external memory, performs comparably to Ours (IE), which combines internal and external memory. This suggests that external memory alone is sufficient and that the policy can adaptively leverage available memory sources.

