# OpenReview forum: "Recurrent model for Sequential reasoning"
_ICLR.cc/2026/Conference — Submitted to ICLR 2026_

### Official Review · Reviewer_7Mq1 · 2025-10-25

**Soundness:** 1
**Presentation:** 1
**Contribution:** 1
**Rating:** 0
**Confidence:** 5

**Summary:**

The paper aims to address the challenge of real-time control and planning, where inputs arrive continuously and decisions must be made asynchronously while planning. The authors consider a two time scales process where planning (reasoning) steps are performed faster than input arrival rate.
They propose a recurrent architecture and an auxiliary memory module.
In their experiments, they compare the proposed architecture against several sequence model architectures on two benchmarks: MiniGrid, and a maze environment.

**Strengths:**

- The problem of effective "real-time" control (and planning) is interesting and important for the community (and has been studied extensively in the literature).
- The proposed method is original.

**Weaknesses:**

`W1`: Throughout the paper there are many inaccuracies and lack of precision, suggesting a failure to identify precisely what is the problem to be addressed, what are the relevant baselines and methods in current literature for that particular problem and setting, and what is the appropriate experimental setup for proper evaluation (for showing progress).
As an example (line 13):
> However, these models are not designed to accept sequential and dynamical input, while many real-world applications as well as the survival of organisms in nature require real-time input to be processed in parallel with the reasoning.

The referred approaches are by definition sequence models, and are designed to accept sequential input, e.g., token sequences.
If I understand correctly, the authors' intention was that these methods are not design for a rather practical setting where there is an active input stream, with inputs arriving at a given rate, while the model is required to provide outputs asynchronously, as a real-time controller, while planning (reasoning).

Importantly, this problem and setting is very different from that of chain-of-thought (CoT), and in fact I would not consider CoT as part of the most relevant literature.
Moreover, the term "reasoning" is typically used in current literature for language-based planning. Here, there is no use of language modality, and thus "reasoning" is not the appropriate term here, perhaps "Planning" is more appropriate.

I refer the authors to the control and reinforcement learning literature, where such settings were extensively studied.
For example, in autonomous driving [1], algorithms perform concurrent planning and control in real-world settings successfully.

[1] Paden, B., Čáp, M., Yong, S. Z., Yershov, D., & Frazzoli, E. (2016). A survey of motion planning and control techniques for self-driving urban vehicles. IEEE Transactions on intelligent vehicles, 1(1), 33-55.




`W2`: The paper contains many unjustified claims and statements.
Example:
> Indeed, a realtime agent, such as a living organism in nature, is required to solve the reasoning task with this deadline constraint.

This is not a trivial claim, please provide supporting evidence (e.g., refer to prior works that validate this claim).





`W3`: It is unclear that effective real-time control can not be achieved by existing methods under various reasonable choices, i.e., that there is a real problem with existing methods.
The authors are responsible for establishing that there is a problem with existing methods in this setting for an important subset of tasks, under certain hardware choices, etc.
Currently, the paper does not convincingly support this claim.

For example, many existing robotic applications operate in real-world setting, in real time, and perform very well [1][2], without the method proposed in this paper. Where is the problem?

[1] Tang, C., Abbatematteo, B., Hu, J., Chandra, R., Martín-Martín, R., & Stone, P. (2025, April). Deep reinforcement learning for robotics: A survey of real-world successes. In Proceedings of the AAAI Conference on Artificial Intelligence (Vol. 39, No. 27, pp. 28694-28698).

[2] Paden, B., Čáp, M., Yong, S. Z., Yershov, D., & Frazzoli, E. (2016). A survey of motion planning and control techniques for self-driving urban vehicles. IEEE Transactions on intelligent vehicles, 1(1), 33-55.





`W4`:  Poor experimental design.
The experimental setup compares *sequence model architectures*. It is thus unclear how the choice of sequence model architecture is relevant for addressing the problem at hand, i.e., real-time asynchronous control.
Real-time response time crucially depends on model size, hardware, architecture choices, and on the difficulty and properties of the specific problem at hand.
The experiments should compare existing methods for real-time control, rather than sequence model architectures. Model architecture, size, hardware, and other factors should be controlled for any real-time control baseline.

In the related works section the authors referred to existing methods. Why are these methods not included in the comparison?

In addition, it is unclear how real-time control was implemented in the experiments, if at all. What is the observation (input) arrival rate? How each baseline processes and computes outputs for such real-time setting? How is the real-time aspect implemented in the experiments? Why and how this design reflects real-world real-time control settings?

**Questions:**

`Q1`: Why is the latent (hidden) dimensionality of the baselines different from that of the proposed method? How this affects the results?

`Q2`: How did you choose the hyperparameters in your experiments? Why?

---

> ### Author Response · Authors · 2025-11-25
>
> Thank you very much for the feedbacks. We respond to your concerns and questions below.
>
> > `W1`: Throughout the paper there are many inaccuracies and lack of precision, suggesting a failure to identify precisely what is the problem to be addressed, what are the relevant baselines and methods in current literature for that particular problem and setting, and what is the appropriate experimental setup for proper evaluation (for showing progress).
>
> We are sorry that our motivation was unclear in the original manuscript. We would like to refer the reviewer to our response to all reviewers.
>
>
> > `W2`: The paper contains many unjustified claims and statements. Example:
> Indeed, a realtime agent, such as a living organism in nature, is required to solve the reasoning task with this deadline constraint.
> This is not a trivial claim, please provide supporting evidence (e.g., refer to prior works that validate this claim).
>
> Would you please elaborate this question further? For this phrase of concern, we wanted to simply raise a point that when a prey is attacked by a predator, the prey must consolidate the strategy of evading before the predator arrives.  We believe that “reasoning” is not officially defined to be in language.
> When we are using the word “reasoning”, we are referring to the process similar to CoT, in which a feature is recursively refined/consolidated  toward the purpose. While CoT’s intermediate outputs are grounded in language, our intermediate outputs are abstract latents.
>
> > `W3`: It is unclear that effective real-time control can not be achieved by existing methods under various reasonable choices, i.e., that there is a real problem with existing methods. The authors are responsible for establishing that there is a problem with existing methods in this setting for an important subset of tasks, under certain hardware choices, etc. Currently, the paper does not convincingly support this claim.
>
> We are sorry that we might have abused the keyword “real-time”.   We did not intend to use this keyword to relate ourselves to the latency problem. Please see our response to all reviewers.
>
> > `W4`: Poor experimental design. The experimental setup compares *sequence model architectures*. It is thus unclear how the choice of sequence model architecture is relevant for addressing the problem at hand, i.e., real-time asynchronous control. Real-time response time crucially depends on model size, hardware, architecture choices, and on the difficulty and properties of the specific problem at hand. The experiments should compare existing methods for real-time control, rather than sequence model architectures.
>
> We would like to emphasize that we are not interested in “real-time asynchronous *control*”.  Although we have never used the term “control” in our manuscript,  we are sorry if there was a misleading expression other than “real-time” we mentioned above.
>
> > In the related works section the authors referred to existing methods. Why are these methods not included in the comparison?
>
> We appreciate your comment regarding the baselines.  As we mention in our response to all reviewers, we added more baselines, such as looped Transformer and CTM; see the table mentioned in the next question.

---

> ### Author Response · Authors · 2025-11-27
>
> > `Q1`: Why is the latent (hidden) dimensionality of the baselines different from that of the proposed method? How this affects the results?
>
> It is a very reasonable concern that, if a model size differs across the baselines,  smaller models may overfit less, enjoying unfair advantage for the OOD generalization. To address this concern, we re-ran the experiments on the maze task after adjusting all baseline models to have a comparable number of parameters to the proposed model.
>
> | Model    | Params. (M) | ID Accuracy (%) | OOD Accuracy (%) |
> |----------|-------------|-----------------|------------------|
> | LSTM     | 2.86        | 52.7 ± 9.6      | 3.3 ± 1.8        |
> | Mamba    | 1.25        | 20.9 ± 0.4      | 19.6 ± 1.9       |
> | TF       | 1.73        | 58.6 ± 4.2      | 10.6 ± 6.1       |
> | Looped TF| 1.73        | 21.4 ± 0.1      | 19.9 ± 0.5       |
> | S5       | 1.64        | 7.6 ± 0.1       | 6.1 ± 0.3        |
> | CTM      | 5.80        | 11.2 ± 0.1      | 7.7 ± 0.4        |
> | Ours     | 1.16        | **96.8** ± 0.1      | **61.2** ± 1.9       |
>
> As shown, our method still performs substantially better in OOD generalization.
>
> In addition, we conducted a new experiment to investigate how the performance of the proposed model changes as we scale its size, where we varied the size by the number of heads and channels in the attention heads of J.  In the newly added Table 6, we evaluated the effect of model size on OOD accuracy in the maze task (3 seeds).
>
> | Channels | Heads | Param. (M) | Mean ± Std (%)  |
> | -------- | ----- | ---------- | ------------ |
> | 64       | 4     | 1.16       | 69.2 ± 3.0 |
> | 128      | 4     | 3.35       | 67.0 ± 4.5 |
> | 256      | 8     | 11.39      | 72.7 ± 2.1 |
>
> The results 4 indicate that OOD performance generally improves as the model size increases. This suggests that the strong OOD performance of the proposed model is not simply due to its relatively small size.
>
> > `Q2`: How did you choose the hyperparameters in your experiments? Why?
>
> For the maze and the (newly added) Dyck tasks, we carefully tuned the learning rate parameter based on validation performance.  For all baseline models of RL, we used the hyperparameters that were publicized on [https://github.com/SafeRL-Lab/BenchNetRL](https://github.com/SafeRL-Lab/BenchNetRL).

---

### Official Review · Reviewer_Cbwp · 2025-10-31

**Soundness:** 1
**Presentation:** 1
**Contribution:** 3
**Rating:** 2
**Confidence:** 4

**Summary:**

This paper proposes a novel recurrent reasoning architecture which works on two timescales; a fast recurrent inner loop and a slower observation timescale to handle real-time streaming inputs. In essence, their method performs many recurrent hidden state updates between each new observation which is received. The authors base their framework off of a Kuramoto oscillatory-based mode (AKOrN). The authors evaluate their approach on synthetic maze path prediction tasks as well as several minigrid RL tasks. Their results show good OOD generalization for these tasks compared to other recurrent models such as LSTM, Mamba-2 and Transformer-XL

I think this paper brings forth interesting ideas, but it is not yet up to the standards of an ICLR main conference track submission. The writing is poor, the experimental results, although interesting, lack depth both in the baselines and benchmarks considered. Although the main ideas behind this paper are interesting, I think it still needs some work. See below for suggestions on how to address these weaknesses.

**Strengths:**

- The architecture proposed by the authors is novel and interesting. It is grounded in observations from the neuroscience/biology literature as well as observations from the LLM reasoning/test time inference literature.
- I think the idea of scaling "fast inner loop" iterations to improve reasoning is interesting and could lead to improved model design.
- The method proposed in the paper has strong OOD generalization on the considered tasks.

**Weaknesses:**

**Writing Style of the paper is poor and needs polishing**

- I find the writing style of the paper to be poor; it feels like this paper was written in a rush. The paper has many typos and poorly constructed sentences. See Questions section for a couple examples. I did not write up an exhaustive list.
- I also feel the narrative of the paper lacks structure. What is the problem you are trying to solve? How are you solving it?
- The paper discusses test time scaling, large reasoning models (LRMs)and continuous CoT a lot as motivations. The authors should make clear that their work is *inspired* by claims from this literature but not a direct extension of it.

**Experiments lack breadth both on the benchmarks/tasks considered and on the model baselines considered**

- 3 seeds is very few for RL. I find that the results lack statistical significance because of this
- I find the experimental section to lack comparable benchmarks. The authors only evaluate on two, mainly synthetic tasks, and do not evaluate on any known tasks from the recurrent model literature such as tasks from the LongRangeArena.
- The baselines (LSTM, Mamba-2, Transformer-XL) are relevant, but the parameter counts shown in Table 4 indicate major capacity mismatches (ours reported at 1.16M / 1.19M parameters vs baselines up to ≈27M). This raises a concern that improvements might partially come from better inductive bias rather than a fair capacity/compute comparison. A fairer comparison requires (a) matching parameter budgets or (b) showing that the gains persist when baselines are scaled to similar parameter counts and compute budgets.
- Important related system baselines that deal with streaming or latent recurrent state (Perceiver variants, Looped Transformer, Universal Transformer with recurrence, stronger RL baselines) should be included or directly discussed in more depth. It could also be interesting to test S4/S5 state space models as they are known to perform better than Mambas on tasks similar to the "Maze" task considered (I am thinking about PathX and related tasks)
- It seems from the ablations done in Fig. 4 that auxiliary memory has a significant impact on the performance of the author's method on the Maze task. The authors should also evaluate other architectures with augmented memory to control for the effect of their architecture

I want to stress that my point is not that the authors should compare against *all* baselines I am proposing here and that they should have experiments on a large set of benchmarks. I am aware this is a methods paper proposing a novel architecture, and thus scaling of the method/adaptation of the method to many different regimes/tasks is not the main focus here. My point is mainly that I do not find the tasks, nor the baseline models to be carefully chosen to properly illustrate the point the authors make. I believe that strengthening the experimental design of the paper could lead to a much stronger submission.

**Questions:**

**Typos**
- Line 107 "we propose to approach realtime sequential reasoning task" not a grammatically correct sentence
- Line 162 "We choose the AKOrN model for the recurrent update because for several reasons."
- There are many more instances of grammatically incorrect sentences, but I will not list all of them.

**Experiments**
- Why did the authors use training curves (with steps) in Figure 3 to support their claim of improved model performance? The claim you are making does not hinge in any way on the training dynamics and you do not discuss this in the surrounding text.
- Why did the authors use the same epoch number, learning rate and batch size across all models? Was the hyperparameter tuning done independently for each considered architecture? Are the authors confident that each baseline model was trained using the best hyperparameter configuration?

**Other**
- A key contributions paragraph/bullet point list could be beneficial to readers and help them extract the claims being made by the paper
- A few references to very recent/arXiv works are used as anchors for claims (e.g., Miyato et al., 2025 AKOrN). Ensure that the relationship between those works and your contribution is explicitly clarified: what is new here vs what AKOrN already provided?
- In Section 3, make explicit how observation encoding delays (encoder cost) factor into the real-time deadline (Sec.2). Clarify whether encoders run within the slow tick or concurrently.
- Expand captions of Figures 3–7 with clearer description of plotted quantities (exact metrics, units).
- Increase the number of seeds shown in the Appendix for reproducibility.
- In Table 1 what do the checkmarks and crosses mean quantitatively. This is explained nowhere

---

> ### Author Response · Authors · 2025-11-25
>
> Thank you very much for the constructive feedbacks. We respond to your concerns and questions below.
>
> > I find the writing style of the paper to be poor; it feels like this paper was written in a rush...
>
>   We were sorry for the initial writing quality of our manuscript. We have revised the introduction to better convey our motivation, and fixed the typographies throughout.  Please see our response to all reviewers.
>
> > I also feel the narrative of the paper lacks structure. What is the problem you are trying to solve? How are you solving it?
>
>   As we state in our response to all reviewers, the main motivation of this work is to extend the capability of a recurrent model to the OOD generalization ability on the sequential tasks. Please see the first section of our response to all reviewers. We succeed in developing this extension by aligning the process of recurrence and the streaming input in the “fast-slow” framework, in which the fast-progressing recurrent loops iteratively consolidate the information of the slow-arriving sequential input.  We find AKOrN to be the choice of the recurrent architecture that best achieves this objective; please see the table below for the ablation study on the maze task.
>
> | Fast Module    | ID Accuracy (%)   | OOD Accuracy (%)   |
> |----------------|-------------------|--------------------|
> | LSTM           | 8.2 ± 1.1         | 4.9 ± 2.2          |
> | TF             | 13.8 ± 0.2        | 10.7 ± 0.9         |
> | AKOrN          | **96.8** ± 0.1        | **61.2** ± 1.9         |
>
> We demonstrate quantitatively with experiments, and also qualitatively analyze how the latent process is naturally learning the structure of the task (Please see the other parts of the response to all reviewers as well).
>
> We revised our manuscript so that it better conveys the problem we are trying to solve and how we are solving it.
>
>
> > The paper discusses test time scaling, large reasoning models (LRMs) and continuous CoT a lot as motivations. The authors should make clear that their work is *inspired* by claims from this literature but not a direct extension of it.
>
> We are sorry that CoT and LRM were overstressed in the original manuscript; we revised the manuscript to convey that these works are inspirations.
>
> > 3 seeds is very few for RL. I find that the results lack statistical significance because of this
>
> We are conducting RL experiments with additional seeds (5 seeds in total). Due to the resource limitation, only the LavaCrossing task has been completed; we will report the full results when the remaining tasks are finished. We also added the training curves with confidence intervals in Figure 11.
>
> > I find the experimental section to lack comparable benchmarks. The authors only evaluate on two, mainly synthetic tasks, and do not evaluate on any known tasks from the recurrent model literature...
>
> To improve your confidence in our work, we also conducted additional experiments on the Dyck task, which is extensively used in the literature. We confirm that our model scales significantly better on OOD domains in terms of the length extrapolation ability; please see the table below, which reports the OOD accuracy on various sequential length ranges from 1 to 2560.
>
> | Model | 1-10 | 10-20 | 20-40 | 40-80 | 80-160 | 160-320 | 320-640 | 640-1280 | 1280-2560 |
> |-------|------|------|------|------|------|------|------|------|------|
> | LSTM | 100.0 ± 0.0 | 95.6 ± 0.7 | 53.6 ± 0.8 | 50.0 ± 0.0 | 50.0 ± 0.0 | 50.0 ± 0.0 | 50.0 ± 0.0 | 50.0 ± 0.0 | 50.0 ± 0.0 |
> | Mamba2 | 100.0 ± 0.0 | 100.0 ± 0.0 | 91.3 ± 6.8 | 61.2 ± 11.7 | 50.7 ± 0.7 | 50.1 ± 0.1 | 50.1 ± 0.1 | 50.1 ± 0.2 | 50.1 ± 0.1 |
> | Transformer | 100.0 ± 0.0 | 99.8 ± 0.1 | 86.6 ± 1.8 | 64.8 ± 1.8 | 51.0 ± 0.4 | 50.0 ± 0.0 | 50.0 ± 0.0 | 50.0 ± 0.0 | 50.0 ± 0.0 |
> | Ours | 100.0 ± 0.0 | 100.0 ± 0.0 | 100.0 ± 0.0 | 100.0 ± 0.0 | 99.9 ± 0.1 | 99.7 ± 0.5 | 99.0 ± 1.3 | 97.8 ± 2.9 | 95.9 ± 5.1 |
>
> We also added more baselines to our experiments; please see our revised paper.

---

> ### Author Response · Authors · 2025-11-25
>
> > The baselines (LSTM, Mamba-2, Transformer-XL) are relevant, but the parameter counts shown in Table 4 indicate major capacity mismatches (ours reported at 1.16M / 1.19M parameters vs baselines up to ≈27M). This raises a concern that improvements might partially come from better inductive bias rather than a fair capacity/compute comparison. A fairer comparison requires (a) matching parameter budgets or (b) showing that the gains persist when baselines are scaled to similar parameter counts and compute budgets.
>
>   We agree that this is a natural concern in evaluating our framework. To address this concern, we re-ran the experiments on the maze task after adjusting all baseline models to have a comparable number of parameters to the proposed model.  As shown in Figure 5, our model still performs substantially better in the OOD setting. Below we paste the table version of Figure 5 for your quick reference. Note that the parameter counts were already matched in the RL tasks.
>
> | Model    | Params. (M) | ID Accuracy (%) | OOD Accuracy (%) |
> |----------|-------------|-----------------|------------------|
> | LSTM     | 2.86        | 52.7 ± 9.6      | 3.3 ± 1.8        |
> | Mamba    | 1.25        | 20.9 ± 0.4      | 19.6 ± 1.9       |
> | TF       | 1.73        | 58.6 ± 4.2      | 10.6 ± 6.1       |
> | Looped TF| 1.73        | 21.4 ± 0.1      | 19.9 ± 0.5       |
> | S5       | 1.64        | 7.6 ± 0.1       | 6.1 ± 0.3        |
> | CTM      | 5.80        | 11.2 ± 0.1      | 7.7 ± 0.4        |
> | Ours     | 1.16        | **96.8** ± 0.1      | **61.2** ± 1.9       |
>
> We also investigated the effect of the size of our model on the OOD, and confirmed that our method generally improves with the one having a greater number of parameters, without suffering the case of overfitting.  In Table 6, we evaluated the effect of model size on OOD accuracy in the maze task (over 3 seeds).  ``Heads'' denotes the number of attention heads in $J$.
>
> | Channels | Heads | Param. (M) | Mean ± Std (%)  |
> | -------- | ----- | ---------- | ------------ |
> | 64       | 4     | 1.16       | 69.2 ± 3.0 |
> | 128      | 4     | 3.35       | 67.0 ± 4.5 |
> | 256      | 8     | 11.39      | 72.7 ± 2.1 |
>
> > Important related system baselines that deal with streaming or latent recurrent state should be included or directly discussed in more depth. It could also be interesting to test S4/S5 state space models as they are known to perform better than Mambas on tasks similar to the "Maze" task considered
>
>   Thank you very much for mentioning these works. We added looped Transformer to the baseline, as well as CTM **and S5**. Please refer to the table version of Figure 5 above.
>
> > It seems from the ablations done in Fig. 4 that auxiliary memory has a significant impact on the performance of the author's method on the Maze task. The authors should also evaluate other architectures with augmented memory to control for the effect of their architecture
>
> Adding auxiliary memories of our proposed type to an arbitrary module would involve non-trivial engineering choices. However, we tested adding an LSTM to an external auxiliary LSTM module; please see Figure 16\.  Below we paste the table version of Figure16 for your quick reference, where `LSTM+E` denotes the memory-equipped LSTM.
> We confirmed that this design does not perform as well as ours, possibly because LSTM lacks the ability to consolidate the information along the latent trajectory (Please see our “Why AKOrN based framework?” section of our response to all reviewers. )
>
>
> | Method         | ID Accuracy (%)   | OOD Accuracy (%)   |
> |----------------|-------------------|--------------------|
> | LSTM           | 52.7 ± 9.6        | 3.3 ± 1.8          |
> | LSTM+E         | 34.9 ± 22.4       | 1.9 ± 0.9          |
> | (Ours + E) | 93.9 ± 1.3|	57.8 ± 2.0 |
>
>
> > Why did the authors use training curves (with steps) in Figure 3 to support their claim of improved model performance? The claim you are making does not hinge in any way on the training dynamics and you do not discuss this in the surrounding text.
>
> Thank you very much for the remark. We replaced the results with bar figures.
>
> > Why did the authors use the same epoch number, learning rate and batch size across all models? Was the hyperparameter tuning done independently for each considered architecture? Are the authors confident that each baseline model was trained using the best hyperparameter configuration?
>
> For the maze and the (newly added) Dyck task, we carefully tuned the learning rate parameter based on validation performance.  For all baseline models of RL, we used the hyperparameters that were publicized on [https://github.com/SafeRL-Lab/BenchNetRL](https://github.com/SafeRL-Lab/BenchNetRL).

---

> > ### Author Response · Authors · 2025-11-26
> >
> > > A few references to very recent/arXiv works are used as anchors for claims (e.g., Miyato et al., 2025 AKOrN). Ensure that the relationship between those works and your contribution is explicitly clarified: what is new here vs what AKOrN already provided?
> >
> > AKOrN is a framework that uses a generalization of Kuramoto model to synchronize the signals from the static input through a recurrent loop to obtain a useful feature, and it is known for its ability to improve the feature by looping more (test time scaling). Geiping et al. also provides a similar ability.  These methods, on the other hand,  are not designed to accept a streaming input. We made this possible by aligning the fast recurrent loop with slow stream, as well as by adding memory mechanisms. We revised our manuscript to state this clearly, especially in Introduction.
> >
> > > In Section 3, make explicit how observation encoding delays (encoder cost) factor into the real-time deadline (Sec.2). Clarify whether encoders run within the slow tick or concurrently.
> >
> >   We apologize we abused the word “real-time” in our work.  We did not intend to use “real-time” to relate to the problem of latency. Please see our first response in this rebuttal. Please also see our response to your question in regards to parameter counts (we provide our evaluation of computational cost)
> >
> > > Expand captions of Figures 3–7 with clearer description of plotted quantities (exact metrics, units).
> >
> >   We are sorry if some parts of our Figures are illegible. We changed the figure sizes and increased our contents in the caption to further elaborate the plot contents.
> >
> > > Increase the number of seeds shown in the Appendix for reproducibility.
> >
> >   For RL, we are running additional experiments to increase the number of seeds from 3 to 5\.  Please see the newly added Figure 11 for the result of LavaCrossing with more seeds.   We will update the results as soon as all the experiments are finished.
> >
> > > In Table 1 what do the checkmarks and crosses mean quantitatively. This is explained nowhere
> >
> > If there is a checkmark for the row entry (configuration name)  on the column of ID, it means that the given configuration is an in-distribution dataset, that is, that the given configuration was used in the training. If there is a checkmark on the the column of OOD, it means that the configuration is OOD (not used for the training)
> >
> > For example, for the Multiroom environment ,   MiniGrid-MultiRoom-N2-S4-v0  was used for the training, and we used MiniGrid-MultiRoom-N4-S5-v0 as well as  MiniGrid-MultiRoom-N6-v0  for OOD evaluation.

---

### Official Review · Reviewer_kmRT · 2025-11-01

**Soundness:** 2
**Presentation:** 3
**Contribution:** 3
**Rating:** 4
**Confidence:** 4

**Summary:**

This paper presents a method that extends the test-time scaling capability of a recurrent model to consider real-time sequential inputs. The proposed method decomposes the system into fast and slow processes. The slow process takes in the observation, and the fast process reasons while waiting for the observations. This paper adopts a recent Kuramoto oscillatory-based model, AKOrN, to update the states in the fast process. It also introduces auxiliary internal and external memory to retain information in fast and slow time scale. The experiment on the egocentric maze and MiniGrid tasks showed that the proposed method improves the performance and the accuracy in MiniGrid also scales as reasoning iteration increases.

**Strengths:**

- This method is inspired by biology and is capable of performing real-time sequential reasoning.
- The proposed method can potentially apply to different latent recurrent reasoning models.
- The experiment shows that the proposed method can improve tasks that require sequential inputs.

**Weaknesses:**

- The main architectural change is the fast and slow recurrent process. How is this idea different from having hierarchical LSTMs where the LSTMs at different levels operate at different time scales?
- The proposed method is mainly based on AKOrN. It is unclear how much of the improvement is from the choice of the update rule. Will other updating rules show the same effect?
- The experiment results in some RL tasks (e.g., MultiRoom) have similar performance to the baseline models. It is hard to conclude the proposed method has better performance.

**Questions:**

- Why do the internal memory only apply to synthetic tasks and the external memory only apply to RL tasks?
- How many reasoning iterations do both experiments use? How much delay is there when increasing the reasoning iteration? What is a realistic number of reasoning iterations to achieve real-time interactions?

---

> ### Author Response · Authors · 2025-11-25
>
> Thank you very much for the constructive feedback and comments. We respond to your concerns and questions below.
>
> > The main architectural change is the fast and slow recurrent process. How is this idea different from having hierarchical LSTMs where the LSTMs at different levels operate at different time scales?
>
> Thank you very much for the suggestion.  We tested Hierarchical LSTM, which we replaced the "fast module" with LSTM in our fast-slow framework on the maze task, and confirmed that it does not scale well out of distribution; please see the table below.
>
> | Fast Module    | ID Accuracy (%)   | OOD Accuracy (%)   |
> |----------------|-------------------|--------------------|
> | LSTM           | 8.2 ± 1.1         | 4.9 ± 2.2          |
> | TF             | 13.8 ± 0.2        | 10.7 ± 0.9         |
> | AKOrN          | **96.8** ± 0.1        | **61.2** ± 1.9         |
>
> We hypothesize that hierarchical LSTM is not performing as well as ours because it is lacking the “consolidation” mechanism that we mentioned in response to all reviewers (“Why AKOrN based framework”?). Further investigation of this difference is an important future research subject.
>
> > The proposed method is mainly based on AKOrN. It is unclear how much of the improvement is from the choice of the update rule. Will other updating rules show the same effect?
>
> Thank you very much for raising this concern. We tested replacing the update rules in the recurrence with LSTM and looped Transformer (Figure 15; please see the table above), and confirmed that they do not scale as well as our choice (AKOrN). Notable differences between the Looped Transformer and AKOrN instantiations of $F$ include the presence of the anti-symmetric matrix $\Omega$ and the normalization scheme: AKOrN uses an anti-symmetric drift plus a projection onto the unit sphere, whereas looped Transformer relies on standard residual connections with pre-norm. Our ablation indicates that these design choices are important for generalization; however, disentangling which component (e.g., the anti-symmetric structure versus the specific normalization) has not been done in depth yet, and we are leaving this for future work.
>
> > The experiment results in some RL tasks (e.g., MultiRoom) have similar performance to the baseline models. It is hard to conclude the proposed method has better performance.
>
> We agree that we do not eclipse the rivals drastically on all tasks. However, as in the case of many novel propositions to date,  we are consistently performing better or equally on most domain.  Moreover, for the multiroom environment, we would like to bring to the attention that our approach scales faster than LSTM with respect to tau.  In Figure 11, we also provided the additional results visualized in the form of training curves,  with more seeds.  We will add similar figures for other environments as soon as possible.
>
> > Why do the internal memory only apply to synthetic tasks and the external memory only apply to RL tasks?
>
> Thank you very much for raising this concern. At this point it is uncertain why some tasks prefer one type of memory to another, and we believe that gradient propagates better or worse to the external memory on  some tasks,  At the same time, as we responded in our message to all reviewers, we confirmed that the model with both types of memory simultaneously performs well on the maze task, so that the user does not have to worry about the choice of the memory type. Please see the table below.
>
> | Method         | ID Accuracy (%)   | OOD Accuracy (%)   |
> |----------------|-------------------|--------------------|
> | Ours(V)        | 65.9 ± 17.2       | 38.6 ± 3.4         |
> | Ours(E)        | 93.9 ± 1.3        | 57.8 ± 2.0         |
> | Ours(I)        | 96.8 ± 0.1        | 61.2 ± 1.9         |
> | Ours(IE)       | 95.8 ± 1.5        | 57.9 ± 4.3         |
>
> `Ours(V)`, `Ours(I)`, `Ours(E)`, and `Ours(IE)` respectively denote our model equipped with no memory, internal memory, external memory, and both.

---

> > ### Author Response · Authors · 2025-11-27
> >
> > > How many reasoning iterations do both experiments use? How much delay is there when increasing the reasoning iteration? What is a realistic number of reasoning iterations to achieve real-time interactions?
> >
> > We used the number of iterations T=5 as a default value for all experiments; please see Table 2. We would like to apologize that our abuse of the word “real-time” that might have made our paper sound as if it relates to latency.  As we respond in our message to all reviewers, our goal is to extend the recurrent architectures like AKOrN to achieve the OOD generalization on sequential tasks.
> > However, in order to address the limitation of our work, we also conducted additional RL experiments to measure the computational cost of our inference; please see the table below.
> >
> > | Model       | T | Average inference time (ms) |
> > |-------------|---|-----------------------------|
> > | LSTM        | - | 0.38                        |
> > | Mamba       | - | 1.92                        |
> > | Transformer | - | 2.21                        |
> > | Ours        | 2 | 4.69                        |
> > | Ours        | 5 | 7.93                        |
> > | Ours        | 10| 14.70                       |
> >
> > For the proposed method, the computation time increases linearly with the number of fast inner loops $T$.

---

### Official Review · Reviewer_Kw2M · 2025-11-01

**Soundness:** 3
**Presentation:** 3
**Contribution:** 3
**Rating:** 6
**Confidence:** 4

**Summary:**

The paper introduces a recurrent reasoning model designed to handle streams of incoming information in $\textbf{real time}$ while operating under a time limit. The approach combines a fast inner-loop reasoning process with a slower sequence of external observations, allowing the system to update its understanding continuously.

The method builds on the AKOrN framework—a Kuramoto-based recurrent updater—by adapting it to work across two timescales and improving its stability with additional memory components, either internal (such as GRU or LSTM units) or external.

Experiments on a synthetic maze navigation task and partially observable MiniGrid reinforcement learning environments show that the model performs well on familiar data and generalizes better to new situations. Increasing the number of inner reasoning steps at test time further improves performance, demonstrating both flexibility and robustness compared to standard recurrent and transformer-based baselines.

**Strengths:**

1. The paper clearly formulates the challenge of real-time sequential reasoning under time constraints, introducing a structured fast/slow update framework that links internal reasoning cycles with observation timescales (Eq. 1; Fig. 1, p. 3).

2. The model extends AKOrN’s hyperspherical latent update mechanism (Eqs. 2–3) to handle streaming inputs and introduces auxiliary memory modules—internal (Eq. 4) and external (Eq. 5)—to maintain stability and temporal continuity (pp. 3–4).

3. The authors provide detailed hyperparameter settings (Tables 2–4, pp. 12–13) and an anonymous code repository, supporting transparency and replicability of results.

**Weaknesses:**

1. Although the deadline constraint is well‑stated (p. 2), there is no wall‑clock latency analysis or throughput/compute‑budget evaluation per observation step. The MiniGrid simulator allows pausing between frames; it is unclear whether the method still outperforms baselines under fixed latency budgets or when T (inner iterations) is capped to meet r eal‑time constraints.The paper’s central “real‑time” positioning therefore remains conceptual rather than empirically demonstrated.

2. The evidence is limited to a synthetic ego‑centric Maze classification setup with a fixed output length S=60 (p. 5) and grid‑world RL (MiniGrid). Absent are higher‑dimensional perception or control (e.g., image‑rich or continuous control), where the proposed fast/slow coupling and memory continuity might be stress‑tested more severely. As a result, claims of general dynamic real‑world decision‑making are only partially supported by the current benchmarks.

3. There is little qualitative analysis of failure modes (e.g., specific MiniGrid layouts where the agent fails, types of partial observability that break the approach), nor disaggregation by sub‑skills (memory vs. planning vs. exploration). This limits insight into why the model helps and when it fails. (Figures 6–7 summarize means ± std, but not error taxonomy; pp. 8–9).

**Questions:**

N/A

---

> ### Author Response · Authors · 2025-11-25
>
> Thank you very much for the constructive feedback. We respond to your concerns and questions below.
>
> > 1. Although the deadline constraint is well‑stated (p. 2), there is no wall‑clock latency analysis or throughput/compute‑budget evaluation per observation step. The MiniGrid simulator allows pausing between frames; it is unclear whether the method still outperforms baselines under fixed latency budgets or when T (inner iterations) is capped to meet real‑time constraints. The paper’s central “real‑time” positioning therefore remains conceptual rather than empirically demonstrated.
>
> We apologize that we abused the word “real-time” in our work.  We did not intend to use “real-time” to relate to the problem of latency.   As we responded in our message to all reviewers, the main motivation of this work is to extend the capability of a recurrent model to the OOD generalization ability on tasks involving streaming input. This is the very reason why we focus on tasks like the egocentric maze and RL with varying configurations (and hence the tasks with indefinite size of $\\tau$ )
>
>
>
> > 2. The evidence is limited to a synthetic ego‑centric Maze classification setup with a fixed output length S=60 (p. 5\) and grid‑world RL (MiniGrid). Absent are higher‑dimensional perception or control (e.g., image‑rich or continuous control), where the proposed fast/slow coupling and memory continuity might be stress‑tested more severely. As a result, claims of general dynamic real‑world decision‑making are only partially supported by the current benchmarks.
>
> We are sorry if our original manuscript sounded as if we are making over-claims  regarding “ general dynamic real‑world decision‑making.“   While we believe that we have succeeded in extending the power of looped recurrent architecture to sequential tasks of various types, we do not claim our extension in the current form can scale to any real-world problem, as it would most likely involve domain-specific techniques entangled with the design of the recurrent module. We would like to remove any overstatements if there are any.
>
> To help improve your concern regarding the size of our environment in relation to the tested episode length, we also added additional experiments of Dyck, a highly memory-dependent task that Transformer and Mamba struggle to solve in an out-of-distribution domain.  Please note in our new Figure 7 that our approach extrapolates over a length greater than 1000 on the OOD domain. Below we paste the table version (OOD accuracy only) for your quick reference. The sequential length ranges from 1 to 2560.
>
> | Model | 1-10 | 10-20 | 20-40 | 40-80 | 80-160 | 160-320 | 320-640 | 640-1280 | 1280-2560 |
> |-------|------|------|------|------|------|------|------|------|------|
> | LSTM | 100.0 ± 0.0 | 95.6 ± 0.7 | 53.6 ± 0.8 | 50.0 ± 0.0 | 50.0 ± 0.0 | 50.0 ± 0.0 | 50.0 ± 0.0 | 50.0 ± 0.0 | 50.0 ± 0.0 |
> | Mamba2 | 100.0 ± 0.0 | 100.0 ± 0.0 | 91.3 ± 6.8 | 61.2 ± 11.7 | 50.7 ± 0.7 | 50.1 ± 0.1 | 50.1 ± 0.1 | 50.1 ± 0.2 | 50.1 ± 0.1 |
> | Transformer | 100.0 ± 0.0 | 99.8 ± 0.1 | 86.6 ± 1.8 | 64.8 ± 1.8 | 51.0 ± 0.4 | 50.0 ± 0.0 | 50.0 ± 0.0 | 50.0 ± 0.0 | 50.0 ± 0.0 |
> | Ours | **100.0 ± 0.0** | **100.0 ± 0.0** | **100.0 ± 0.0** | **100.0 ± 0.0** | **99.9 ± 0.1** | **99.7 ± 0.5** | **99.0 ± 1.3** | **97.8 ± 2.9** | **95.9 ± 5.1** |
>
>
> We do agree that, because of the limit of computational resources and time,  we have not stress-tested our approach with super-high-dimensional tasks with a greater number of entities to memorize.  However, we hope that our Dyck experiment can at least demonstrate that we are presenting a framework that can hold onto self-discovered crucial information for an indefinite duration of the episode.
>
> Extending this mechanism to larger-scale problems indeed is a top-priority future research direction.
>
>
> > 3. There is little qualitative analysis of failure modes (e.g., specific MiniGrid layouts where the agent fails, types of partial observability that break the approach), nor disaggregation by sub‑skills (memory vs. planning vs. exploration). This limits insight into why the model helps and when it fails. (Figures 6–7 summarize means ± std, but not error taxonomy; pp. 8–9).
>
> Thank you very much for the suggestion. We agree that analyzing the failure modes would be very valuable.
> At this point, we are aware that, for “a bad choice of seed”, the model sometimes forgets what it has consolidated in the past (maze) or gets caught in a small orbit (RL).
> We will make an effort to further investigate these aspects within the remaining rebuttal period and will include any findings that can further clarify the strengths and limitations of our approach.

---

### Official Review · Reviewer_YVJH · 2025-11-01

**Soundness:** 2
**Presentation:** 2
**Contribution:** 2
**Rating:** 2
**Confidence:** 3

**Summary:**

Authors propose to adopt so-called reasoning module to the realtime sequential reasoning task, in which the agent is required to engage in reasoning and inference at the same time as in-taking the realtime observation sequence. Toward this goal, the authors incorporate the recurrent module into the coupled system of fast and slow processes, in which the module reasons at the fast time scale by sequentially integrating into its state the sensory-observation that occurs at the slow time scale.

**Strengths:**

- Authors show their approach in reinforcement learning tasks and synthetic tasks, and show that it performs competitively against other well-known models for sequential processing, such as LSTM, Mamba, and TransformerXL.

- Authors demonstrate that their strategy enables effective test-time scaling of the recurrent module to the time scale of observation, so that an agent makes better decisions by observing more at the test time; That is, the longer the agent is immersed in the given environment, the deeper the reasoning and, hence, the decision making.

- With this test-time scaling, the proposed approach leads to improved decision making on more complex tasks, surpassing LSTM, Mamba, and TransformerXL on Maze and MiniGrid tasks.

**Weaknesses:**

1) The authors use the term "reasoning," but in the result tables and visualizations of the solved tasks, it appears that they simply generate a sequence of output actions.

2) The authors examined the effectiveness of their approach only on relatively simple maze-solving and MiniGrid tasks; they did not conduct experiments with more complex robotic simulations. The practical usefulness of the developed approach is questionable.

3) Figure 1 is not informative enough. What do the captions in the figure mean? How do the reasoning module and memory module interact?

4) The comparison with basic action generation methods is incomplete; for example, the recent RATE method [1] or the classic Decision Transformer [2] are not mentioned.

[1] Cherepanov, E., Staroverov, A., Yudin, D., Kovalev, A. K., & Panov, A. I. (2023). Recurrent action transformer with memory. arXiv preprint arXiv:2306.09459.

[2] Chen, L., Lu, K., Rajeswaran, A., Lee, K., Grover, A., Laskin, M., ... & Mordatch, I. (2021). Decision transformer: Reinforcement learning via sequence modeling. Advances in neural information processing systems, 34, 15084-15097.

5) The article does not provide information on the inference performance of the developed model, which is critical for fast simulation.

6) The article's formatting should be improved; punctuation marks are missing at the end of the formulas.

**Questions:**

1) In the section on reinforcement learning, the authors write that their model is trained end-to-end using Proximal Policy Optimization. Why was this choice of RL method made?

---

> ### Author Response · Authors · 2025-11-25
>
> Thank you very much for the constructive feedbacks. Below we respond each of tour concern.
>
> > 1. The authors use the term "reasoning," but in the result tables and visualizations of the solved tasks, it appears that they simply generate a sequence of output actions.
>
> We are sorry if there was a miscommunication regarding the terminology of “reasoning”.
> By reasoning we refer to the iterative process of organizing the inputs for the purpose of task solving.  For example, given a static prompt, Chain-of-Though (CoT) iterates the organization of the idea to produce a conclusive remark.  Likewise, the models such as AKOrN iterates the update with the recurrent module to produce a feature, whose form would depend on tasks – it can be a solution of a maze, or it can be a classification label.
> The fundamental difference of these models from CoT is that, at every round of the recursion, the intermediate outputs are not grounded to a fixed set of representations like “word”.
>
> When we extend the recurrent mechanism to accept the streaming input and iteratively produce the output, it will end up in a sequence of output.  For the RL setting, the output was a sequence of actions, and for the maze task, the output was a sequence of the model's refinement for the shortest path solution.
> We would very much appreciateI if there is a reference that states that “reasoning”, by definition,  must be associated to language, so that we will be careful with the terminology more carefully.
>
> > 2. The authors examined the effectiveness of their approach only on relatively simple maze-solving and MiniGrid tasks; they did not conduct experiments with more complex robotic simulations. The practical usefulness of the developed approach is questionable.
>
> We are sorry if our motivation was unclear. We intend to present our work with the purpose of proposing a new way of extending the capability of the recent powerful recurrent architecture to the OOD generalization on sequential tasks.  Just like the works like CTM, HRM and AKOrN, our work is not to present a specific algorithm that is fine tuned for large scale experiments with domain specific knowledge.
>
> In order to improve your confidence in our work, we also conducted additional experiments, including Dyck, a new symbolic task. Please see our response to all reviewers .
>
>
> > 3. Figure 1 is not informative enough. What do the captions in the figure mean? How do the reasoning module and memory module interact?
>
> We are sorry that our figure was not informative. In this figure our intention was to convey how the slow process of the input stream is injected into the fast process in a way similar to input injection (Fan et al., 2025). To clarify our point, we updated Figure 1 to highlight the difference of our framework from the other architectures.
>
> > 4. The comparison with basic action generation methods is incomplete; for example, the recent RATE method \[1\] or the classic Decision Transformer \[2\] are not mentioned.
>
> We would like to note that, by the definition of the task we are tackling (which involves streaming input), our problem in the form of RL is *online*. Decision Transformer is evaluated with *behavior cloning*. RATE is also defined for *offline* setting — extending them to online setting is nontrivial and involves design choices.
> Also, it is to be noted that our method is not specialized for reinforcement learning, and we believe that comparing our method against these methods on a fair ground also requires additional design choices on our side.
>
> > 5. The article does not provide information on the inference performance of the developed model, which is critical for fast simulation.
>
> Although we are uncertain about what is meant by “inference performance” here, we do evaluate our models on the out of distribution dataset. Also, in the revision we reported the actual computational cost on the RL tasks in comparison to other models. Please see the table below. For our model, the computation time increases linearly with the number of fast inner loops $T$ compared with the baselines.
>
> | Model       | T | Average inference time (ms) |
> |-------------|---|-----------------------------|
> | LSTM        | - | 0.38                        |
> | Mamba       | - | 1.92                        |
> | Transformer | - | 2.21                        |
> | Ours        | 2 | 4.69                        |
> | Ours        | 5 | 7.93                        |
> | Ours        | 10| 14.70                       |
>
> We are sorry if our motive was not clear – we would like to emphasize that our goal is not to solve a “real time control” problem. Our goal is to extend the recurrent architectures like AKOrN to achieve the OOD generalization on sequential tasks.  We would like to refer to our response to all reviewers.

---

> > ### Author Response · Authors · 2025-11-27
> >
> > > 6. The article's formatting should be improved; punctuation marks are missing at the end of the formulas.
> >
> > We are sorry for the formatting errors, we will fix these errors in the revision.
> >
> > >  Question 1. In the section on reinforcement learning, the authors write that their model is trained end-to-end using Proximal Policy Optimization. Why was this choice of RL method made?
> >
> > We chose PPO simply as a popular choice of reinforcement learning technique.  In order to validate if our works’ efficacy is dependent on a specific task-method pair, we conducted our experiments on a range of tasks, including maze and, additionally, Dyck.  Please see our response to all reviewers in regard to additional experiments.

---

### Official Review · Reviewer_dSwD · 2025-11-01

**Soundness:** 2
**Presentation:** 3
**Contribution:** 2
**Rating:** 2
**Confidence:** 2

**Summary:**

The paper studies the problem of real-time sequential reasoning: settings where observations arrive as a stream and the agent must keep reasoning while new inputs are arriving, as opposed to ingesting a fixed context and only then iterating. The authors propose a two-timescale architecture that couples a slow observation process (new sensory input every data step \tau) with a fast recurrent reasoning process that can run multiple inner updates (T steps) per observation. Specifically, at each slow data step the observation is encoded into a context vector C_\tau, and a fast recurrent core integrates this new context into its hidden state by iterating a shared update multiple times. The paper also introduces auxiliary memory to stabilise long sequences and prevent representational collapse. The method is evaluated on (i) an ego-centric Maze task, where the model must infer a global path from local views over time, and (ii) MiniGrid RL tasks in a zero-shot OOD setup. In both, the proposed model matches or outperforms LSTM, Mamba-2, and Transformer-XL, especially on OOD mazes and longer-horizon RL variants. The key empirical claim is that success improves as we allow more inner recurrent steps at test time (i.e. test-time scaling works even when inputs are streaming) unlike many previous latent-recurrent approaches that assumed a fixed input window.

**Strengths:**

- The split between a slow observation process and a fast reasoning loop is intuitive and well explained.
- The test environments (maze and MiniGrid) are a simple yet reasonable testbed for partial observability and sequential reasoning.
- The ablation shows that auxiliary memory improves OOD maze performance.

**Weaknesses:**

- The proposed model performs several internal reasoning steps for each observation, while all baselines do only one. Does this mean it effectively gets more compute per timestep? If so, the performance gains might partly come from having more compute rather than from the architecture itself.
- The paper uses an internal auxiliary memory for the maze task and an external one for the RL tasks. There’s no general rule for when to use which, which suggests that the design might need to be adjusted for each domain rather than being a single unified approach.
- In the maze task, the baselines are much larger (up to 25–27M parameters) compared to the proposed model (1.1M). Given the small dataset, this could make the baselines more prone to overfitting.

**Questions:**

- Have you tried training with T reasoning steps and evaluating with T’ > T on the same input sequence?
- What guided the choice between the internal and external memory versions? Could they be combined or selected automatically?
- What happens if the AKOrN module is replaced with a simpler recurrent unit?
- In the maze task, did you explore smaller baselines to check if overfitting explains the gap in OOD performance?

---

> ### Author Response · Authors · 2025-11-25
>
> Thank you very much for constructive feedback.
>
> > The proposed model performs several internal reasoning steps for each observation, while all baselines do only one. Does this mean it effectively gets more compute per timestep? If so, the performance gains might partly come from having more compute rather than from the architecture itself.
>
> We agree that this is a legitimate concern in practice. To investigate whether the additional compute is not the sole source of the performance gain, we conducted an ablation study in which we replaced the “fast” module of our framework with different baseline models. As shown in the table below, even though all methods perform the same  **$T$ internal inference steps** (i.e., they all loop for $T$ reasoning steps), our proposed model (AKOrN) still achieves substantially higher performance than any of these variants.
>
> | Fast Module    | ID Accuracy (%)   | OOD Accuracy (%)   |
> |----------------|-------------------|--------------------|
> | LSTM           | 8.2 ± 1.1         | 4.9 ± 2.2          |
> | TF             | 13.8 ± 0.2        | 10.7 ± 0.9         |
> | AKOrN          | **96.8** ± 0.1        | **61.2** ± 1.9         |
>
> This suggests that the observed gains cannot be explained by increased compute (larger $T$) alone, and that the architectural changes introduced by AKOrN are fundamentally important. Table 7 also shows that, with the AKOrN fast module, the OOD accuracy improves monotonically as the number of training-time steps $T$ increases. We also paste below the copy of Table 7 (3 seeds).
>
> | T  | Mean ± Std (%)   |
> | -- | ------------- |
> | 1  | 30.6 ± 13.5 |
> | 2  | 43.1 ± 18.6 |
> | 4  | 59.5 ± 3.6  |
> | 8  | 67.3 ± 0.6  |
> | 16 | 67.3 ± 3.2  |
>
> > The paper uses an internal auxiliary memory for the maze task and an external one for the RL tasks. There’s no general rule for when to use which, which suggests that the design might need to be adjusted for each domain rather than being a single unified approach.
> > What guided the choice between the internal and external memory versions? Could they be combined or selected automatically?
>
> We agree that this can be of concern in the application. We conducted an extensive ablation study on the maze task, comparing four memory configurations: no memory, internal memory only, external memory only, and both internal and external memory. We report the results below.
>
> | Method         | ID Accuracy (%)   | OOD Accuracy (%)   |
> |----------------|-------------------|--------------------|
> | Ours(V)        | 65.9 ± 17.2       | 38.6 ± 3.4         |
> | Ours(E)        | 93.9 ± 1.3        | 57.8 ± 2.0         |
> | Ours(I)        | 96.8 ± 0.1        | 61.2 ± 1.9         |
> | Ours(IE)       | 95.8 ± 1.5        | 57.9 ± 4.3         |
>
> `Ours(V)`, `Ours(I)`, `Ours(E)`, and `Ours(IE)` respectively denote our model equipped with no memory, internal memory, external memory, and both. The results show that the internal-only configuration performs slightly better, but the difference compared to using both memories is negligible.  Our results suggest that using both types of memory does not cause negative interference between them.
> As a general rule, we therefore suggest using both internal and external memory.

---

> ### Author Response · Authors · 2025-11-26
>
> > In the maze task, the baselines are much larger (up to 25–27M parameters) compared to the proposed model (1.1M). Given the small dataset, this could make the baselines more prone to overfitting.
> > In the maze task, did you explore smaller baselines to check if overfitting explains the gap in OOD performance?
>
> We agree that this is a natural concern in evaluating our framework. To address this concern, we re-ran the experiments on the maze task after adjusting all baseline models to have a comparable number of parameters to the proposed model.  As shown in the table below, our model still performs substantially better in the OOD setting.
>
> | Model    | Params. (M) | ID Accuracy (%) | OOD Accuracy (%) |
> |----------|-------------|-----------------|------------------|
> | LSTM     | 2.86        | 52.7 ± 9.6      | 3.3 ± 1.8        |
> | Mamba    | 1.25        | 20.9 ± 0.4      | 19.6 ± 1.9       |
> | TF       | 1.73        | 58.6 ± 4.2      | 10.6 ± 6.1       |
> | Looped TF| 1.73        | 21.4 ± 0.1      | 19.9 ± 0.5       |
> | S5       | 1.64        | 7.6 ± 0.1       | 6.1 ± 0.3        |
> | CTM      | 5.80        | 11.2 ± 0.1      | 7.7 ± 0.4        |
> | Ours     | 1.16        | **96.8** ± 0.1      | **61.2** ± 1.9       |
>
> In addition, we conducted a new experiment to investigate how the performance of the proposed model changes as we scale its size, where we varied the size by the number of heads and channels in the attention heads of J. The results in Table 6 indicate that OOD performance generally improves as the model size increases. This suggests that the strong OOD performance of the proposed model is not simply due to its relatively small size. We also paste below the copy of Table 6, computed on the maze task. Reported values are OOD accuracies, and  ``Heads'' denotes the number of attention heads in $J$.
>
> | Channels | Heads | Param. (M) | Mean ± Std (%)  |
> | -------- | ----- | ---------- | ------------ |
> | 64       | 4     | 1.16       | 69.2 ± 3.0 |
> | 128      | 4     | 3.35       | 67.0 ± 4.5 |
> | 256      | 8     | 11.39      | 72.7 ± 2.1 |
>
> > Have you tried training with T reasoning steps and evaluating with T’ \> T on the same input sequence?
>
> Unfortunately, unlike for the time scale of tau (slow),  we were not able to confirm test-time scaling with respect to T in the RL setting. We added this ablation in Figure 14\ (see the table below).
>
> | T  | Accuracy (%) |
> | -- | ------------ |
> | 2  | 11.0         |
> | 5  | 31.5         |
> | 8  | 18.2         |
> | 10 | 6.3          |
>
> However,  we still maintain that we can improve the performance by increasing T in the training (Table 7, mentioned in our first response).
>
> > What happens if the AKOrN module is replaced with a simpler recurrent unit?
>
> As we explained above, we confirmed that replacing AKOrN fast process with LSTM or Transformer does not perform as well as ours in OOD generalization. We report the accuracy on the maze task below.
>
> | Fast Module    | ID Accuracy (%)   | OOD Accuracy (%)   |
> |----------------|-------------------|--------------------|
> | LSTM           | 8.2 ± 1.1         | 4.9 ± 2.2          |
> | TF             | 13.8 ± 0.2        | 10.7 ± 0.9         |
> | AKOrN          | **96.8** ± 0.1        | **61.2** ± 1.9         |
>
> It is probable that AKOrN’s consolidation ability, that we mention in both the response to all reviewers and in our revision (Figures 7, 10), is working in favor of organizing the information. Further investigation of this mechanism is an important future research subject.

---

### Official Review · Reviewer_Awzc · 2025-11-02

**Soundness:** 1
**Presentation:** 2
**Contribution:** 2
**Rating:** 2
**Confidence:** 2

**Summary:**

This paper addresses real-time sequential reasoning where a model must reason while observations arrive as a stream, rather than processing a complete sequence first. The authors propose coupling a fast recurrent reasoning module (AKOrN) with slow observational dynamics, using auxiliary memory to maintain continuity during streaming inputs.

**Strengths:**

- Real-time sequential reasoning is genuinely underexplored. The deadline constraint formalization is precise, and the biological grounding provides solid analogous motivation for the approach.
- Using AKOrN (Kuramoto oscillators on hyperspheres) adds mathematical rigor beyond standard RNNs. The fast-slow timescale separation is architecturally elegant. This method has potential when tested with further experiments and evaluations.

**Weaknesses:**

- The method shows better test-time scaling on LavaCrossing and MultiRoom but not DoorKey. The authors' explanation (task-dependent benefits) is reasonable but raises a critical question: how do we know a priori which tasks will benefit? Without a principled framework for predicting when this approach helps, practitioners can't know whether to use it. The paper needs either: (a) a theoretical characterization of suitable task types, or (b) extensive empirical analysis across diverse task categories.
- Only toy MiniGrid World environments low dimensional observations and discrete actions are tested. No validation on complex fields like robotics, vision tasks, language, or any realistic sequential decision-making. The strong maze results are promising, but one synthetic task is insufficient to claim this is a general approach for "real-time sequential reasoning."
- The method requires much more computation than baseline through iterative forward passes. However, there is zero analysis of compute cost or scalability. For real-time applications, efficiency is core to the problem definition. The method's ability for "real time" sequential reasoning requires further justification.

**Questions:**

- The paper cites COCONUT, Looped Transformer, and CTM but doesn't compare against them experimentally. Can you justify for this?
- You use sophisticated Kuramoto dynamics on hyperspheres, but provide no ablations. Can you provide more analysis to show that this is essential?

---

> ### Author Response · Authors · 2025-11-25
>
> Thank you very much for constructive feedback.
>
> > The method shows better test-time scaling on LavaCrossing and MultiRoom but not DoorKey. The authors' explanation (task-dependent benefits) is reasonable but raises a critical question: how do we know a priori which tasks will benefit? Without a principled framework for predicting when this approach helps, practitioners can't know whether to use it. The paper needs either: (a) a theoretical characterization of suitable task types, or (b) extensive empirical analysis across diverse task categories.
>
> Doorkey is somewhat special because, by wandering aimlessly in the environment long enough, one can eventually find the key, open the door and reach the goal by chance. That is, unlike LavaCrossing, it is a task in which an agent with a bad policy can leverage the law of large numbers to succeed. It may be the case that the policy learned by our approach in small maps (5x5)  tends to have a smaller component of “random exploration”.
>
> To help resolve your concern (b), we also conducted additional sets of experiments as well, including the evaluation on a new Dyck task; please see our response to all reviewers. On the Dyck task in particular, our approach scales significantly better than the Transformer and Mamba. Below, we report the OOD accuracy for your quick reference. The sequential length ranges from 1 to 2560.
>
> | Model | 1-10 | 10-20 | 20-40 | 40-80 | 80-160 | 160-320 | 320-640 | 640-1280 | 1280-2560 |
> |-------|------|------|------|------|------|------|------|------|------|
> | LSTM | 100.0 ± 0.0 | 95.6 ± 0.7 | 53.6 ± 0.8 | 50.0 ± 0.0 | 50.0 ± 0.0 | 50.0 ± 0.0 | 50.0 ± 0.0 | 50.0 ± 0.0 | 50.0 ± 0.0 |
> | Mamba2 | 100.0 ± 0.0 | 100.0 ± 0.0 | 91.3 ± 6.8 | 61.2 ± 11.7 | 50.7 ± 0.7 | 50.1 ± 0.1 | 50.1 ± 0.1 | 50.1 ± 0.2 | 50.1 ± 0.1 |
> | Transformer | 100.0 ± 0.0 | 99.8 ± 0.1 | 86.6 ± 1.8 | 64.8 ± 1.8 | 51.0 ± 0.4 | 50.0 ± 0.0 | 50.0 ± 0.0 | 50.0 ± 0.0 | 50.0 ± 0.0 |
> | Ours | **100.0 ± 0.0** | **100.0 ± 0.0** | **100.0 ± 0.0** | **100.0 ± 0.0** | **99.9 ± 0.1** | **99.7 ± 0.5** | **99.0 ± 1.3** | **97.8 ± 2.9** | **95.9 ± 5.1** |
>
> > Only toy MiniGrid World environments low dimensional observations and discrete actions are tested. No validation on complex fields like robotics, vision tasks, language, or any realistic sequential decision-making. The strong maze results are promising, but one synthetic task is insufficient to claim this is a general approach for "real-time sequential reasoning."
>
> We would like to emphasize that, as we state in our response to all reviewers, we abused the term “real-time” in the original manuscript, and that our aim is to solve a sequential inference problem of indefinitely long length under the more general “causal constraint” by extending the recurrent models. Please also see our revised section 2\.
> That being said, we hope that our additional sets of experiments and ablation studies can improve your confidence in our work.  Please see our response to all reviewers.
>
> We hope to ask for your understanding of our intention to present this study as a proposition of a new way of extending the OOD capability of recurrent models, and we believe that scaling our framework to a more realistic setting is future work.

---

> ### Author Response · Authors · 2025-11-25
>
> > The method requires much more computation than baseline through iterative forward passes. However, there is zero analysis of compute cost or scalability. For real-time applications, efficiency is core to the problem definition. The method's ability for "real time" sequential reasoning requires further justification.
>
> It is true that, because of the loop in the fast process, our models require much more computation time in comparison to Mamba and Transformer (\>3 times in RL). Below, we report the forward computation wall-clock time per batch in the Minigrid task. For the proposed method, the computation time increases linearly with the number of fast inner loops $T$ compared with the baselines.
>
> | Method      | T | Average inference time (ms) |
> |-------------|---|-----------------------------|
> | LSTM        | - | 0.38                        |
> | Mamba       | - | 1.92                        |
> | Transformer | - | 2.21                        |
> | Ours        | 2 | 4.69                        |
> | Ours        | 5 | 7.93                        |
> | Ours        | 10| 14.70                       |
>
> We also note that this computational overhead depends in part from software constraints. Competing models e.g. Mamba rely on highly optimized CUDA kernels and well-supported library implementations, whereas our model currently lacks comparable low-level optimizations. We mentioned this limitation at Conclusion in our revised paper.
>
> We would like to clarify again that this fact does not undermine the motivation of our study, and we apologize that we might have abused the terminology “real-time”. In making a reference to this keyword, we were originally trying to convey our motivation of extending the recent recurrent algorithm to the out-of-distribution generalization on the sequential input stream that is processed iteratively along with the recurrence.
>
> Please also see our response to all reviewers.
>
> > The paper cites COCONUT, Looped Transformer, and CTM but doesn't compare against them experimentally. Can you justify for this?
>
> Thank you for pointing this out.  We added looped Transformer and CTM to our baselines for the maze task; please see the result below.
>
> | Method         | ID Accuracy (%)   | OOD Accuracy (%)   |
> |----------------|-------------------|--------------------|
> | LSTM           | 52.7 ± 9.6        | 3.3 ± 1.8          |
> | Mamba          | 20.9 ± 0.4        | 19.6 ± 1.9         |
> | TF             | 58.6 ± 4.2        | 10.6 ± 6.1         |
> | Looped TF      | 21.4 ± 0.1        | 19.9 ± 0.5         |
> | S5             | 7.6 ± 0.1         | 6.1 ± 0.3          |
> | CTM            | 11.2 ± 0.1        | 7.7 ± 0.4          |
> | Ours        | **96.8** ± 0.1        | **61.2** ± 1.9         |
>
> Please note that Coconut is specialized for language modeling (e.g., it “switches back and forth between the ‘language mode’ and ‘latent mode’”); we excluded this work because this method is trained on the dataset of CoT, which is usually unavailable (especially for non-language domain). Please also see our response to all reviewers regarding our added baselines.
>
> > You use sophisticated Kuramoto dynamics on hyperspheres, but provide no ablations. Can you provide more analysis to show that this is essential?
>
> We are sorry that we were not clear enough in our justification of the usage of the Kuramoto model. The mechanism of AKOrN is, as stated in Miyato et al.(2025), grounded in the theory of Kuramoto dynamics on the n-sphere, which has the capability of synchronization. It was our hope that this mechanism would work in favor of synchronizing the internal fast process with the slower input stream.
>
> To validate our intuition, we tracked the energy trajectories and latent space trajectories of our fast process in Dyck task and RL task. We are able to confirm that, unlike SSM, our approach is sequentially consolidating the task-relevant concept like finding key and door in RL, and depth and bracket type in Dyck; see Figures 7,8 and 10\. We also conducted an ablation study and observed that the AKOrN-type fast module generalizes better in the maze task (both OOD and ID) in comparison to LSTM and Transformer-based dynamics; see the result below.
>
> | Module         | ID Accuracy (%)   | OOD Accuracy (%)   |
> |----------------|-------------------|--------------------|
> | LSTM           | 8.2 ± 1.1         | 4.9 ± 2.2          |
> | TF             | 13.8 ± 0.2        | 10.7 ± 0.9         |
> | AKOrN          | **96.8** ± 0.1        | **61.2** ± 1.9         |
>
> Please see our response to all reviewers as well as the new figures in the revision.
>
> We would like to reiterate that, as commented by the reviewer KmRT, the purpose of our paper is to explore the way to extend the capability of recurrent models to the OOD generalization on the task with sequential streaming input. It was essential for us to include in our scope the recurrent model with promising capabilities like AKOrN.

---

### Official Review · Reviewer_7zsu · 2025-11-03

**Soundness:** 3
**Presentation:** 3
**Contribution:** 3
**Rating:** 8
**Confidence:** 3

**Summary:**

The authors of this submission present an approach to incorporate real-time reasoning in recurrent neural networks. Typically, encoder-decoder sequence models ingest the whole input sequence before processing it to generate the output sequence. Here, the authors motivate parallel processing of partial input sequences while continuing to observe the rest of the sequence. The authors motivate this approach via the requirement of real-time reasoning in reinforcement learning and sequential prediction problems. The recently introduced AKOrN network is adapted to update the internal hidden state at a faster rate relative to the input data clock, and two dedicated fast and slow memory banks are integrated into the adapted AKOrN network that further enhance reasoning performance on synthetic problems like Maze solving.

**Strengths:**

Strengths:
- The proposed work is a neat extension of recently proposed AKOrN to enhance its sequential reasoning capabilities. The idea of processing inputs as they arrive in a streaming fashion is widely applicable, and the approach here can potentially extend to real-world applications such as video / audio compression with inherently streaming input characteristics.
- Empirical results show the strength of the proposed approach compared to other baselines in both in- and out-of-distribution settings for challenging synthetic reasoning problems.
- The application to reinforcement learning is also particularly intriguing, I believe the proposed sequential reasoning will be vital to making fast and accurate policies for real world RL.

**Weaknesses:**

Weaknesses:
- While the empirical results are strong, the tasks used to measure performance gains of real-time reasoning are all toy tasks that don't transfer to performance on naturalistic data. It would be great if the authors could comment on how the proposed approach is relevant to SOTA foundation models which also operate on sequences of input.
- The proposed idea of slow and fast reasoning is not necessarily novel. It is an idea that has been explored in key prior work, e.g. Adaptive computation time for recurrent neural networks, by Graves et al., 2017.

**Questions:**

NA. Please refer to my review above.

---

> ### Author Response · Authors · 2025-11-25
>
> Thank you very much for constructive feedback. We agree that the idea of processing inputs as they arrive in a streaming fashion is widely applicable, and that the approach here can potentially extend to real-world applications when complemented with task-specific design.
>
> > While the empirical results are strong, the tasks used to measure performance gains of real-time reasoning are all toy tasks that don't transfer to performance on naturalistic data. It would be great if the authors could comment on how the proposed approach is relevant to SOTA foundation models which also operate on sequences of input.
>
> Thank you very much for your response. Indeed, our ultimate goal is to scale our result to the out of distribution task in the real world environment. However, as is the case with AKOrN and CTM, we are presenting this research to propose a new approach to the reasoning on a sequential stream. As a framework developed from scratch,  we feel that we cannot compare it against the existing SOTA foundation models that are carefully fine-tuned with the art of architectural choices and training procedures.
> Still, given that many foundation models to date are based on the family of SSM and Transformer, we may say that our comparison with Mamba and Transformer type architectures suggest that may further advance the frontier of foundation models by incorporating the mechanism like what we suggest in our work.
>
> Also, in order to improve your confidence in our work, we added additional experiments with more baselines. Please see our response to all reviewers.
>
> > The proposed idea of slow and fast reasoning is not necessarily novel. It is an idea that has been explored in key prior work, e.g. Adaptive computation time for recurrent neural networks, by Graves et al., 2017\.
>
> Indeed, we mention adaptive computation time (ACT) in the related works, as well as other works that use slow vs fast reasoning. However, we are different from ACT in several aspects. Firstly, as we mention in our comparison to looped Transformer in response to all reviewers, our framework leverages the recurrence with the capability of consolidation (Geshkovski et al., 2023), (Miyato et al., 2025\), and we pass-on the consolidated information to the next time step (We also would like to refer the reviewer to our added analysis that aligns with our claim — please see the section of “Why AKOrN-based framework?” in our response to all reviewers.
> Secondly, to maintain the link between fast and slow processes, our framework introduces an additional  memory structure.
> Our contribution is not the invention of the fast-slow mechanism, but rather the way we combine it with modern recurrent mechanism that has the aforementioned capability like consolidation to extend their work to the out of distribution generalization for the sequential input stream,

---

### Author Response · Authors · 2025-11-25
**Response to all reviewers Part1**

We would like to thank all reviewers for constructive feedback. We are sorry that our response came in late; we had **eight** reviewers, and it took us that much longer to craft the responses.
We have incorporated the reviewers’ feedback and revised the paper. The main revisions are highlighted in blue. In this response to all reviewers, we would like to address concerns shared by many reviewers.

The following is the list of the newly added figures and tables:

- Main text: Figures 4, 5, 6, 7, 8, 10, 11\.
- Appendix: Tables 3, 4, 5, 6, 7, 8; Figures 13, 14, 15, 16, 17, 18, 19\.

# Clarification of our motivation

Firstly,  we apologize that our motivations and claims were unclear in the original manuscript, and that we might have caused misunderstanding among some reviewers by **our abuse of the word “real time”**.   As pointed out by the reviewers including kmRT,  the main motivation of this work is to extend the capability of a recurrent model to the OOD generalization ability on the sequential tasks.
This is the very reason why we focus on tasks like egocentric maze and RL with varying configurations.   When we consider the OOD situations that require exploration, the sequence length can grow indefinitely, and there would be a need to consider the input not as a context of a fixed length but as an input stream.
We observed that the extended model succeeded in OOD generalization by using the recurrent model to process (Figures 5, 6, 7, 8, 10, 11\) and **consolidate** the input information at a faster time scale than the input stream, especially when equipped with a memory mechanism (Figures 8, 10).  We revised our introduction to align more closely with our intended motivation and results.
To help resolve the misunderstanding, we also revised Section 2, replaced the \*deadline constraint\* with the  more general *causal constraint*, and  removed the term “real-time” from the manuscript altogether.

---

> ### Author Response · Authors · 2025-11-25
> **Response to all reviewers Part2**
>
> # Additional Experiments
>
> Also there were many comments requesting more diverse experiments with more baselines. In hope to improve your confidence in our work, we conducted additional sets of experiments. Since we received eight reviews and the concerns are diverse, we are still running several additional experiments in hope to directly address the reviewers’ main requests. We will provide the results and analyses for these ongoing experiments as soon as they are completed during the rebuttal period.
>
> ## Dyck Language
>
> In section 5.2, we applied our fast-slow framework with two coupled fast processes to a sequential symbolic task on Dyck language (Hewitt et al., 2020\) with $30$ bracket types and depth $5$. This is a task that requires long term memory of stacked parenthesis.
> As we show in a newly added Figure 7, our approach extrapolates beyond the training sequential length almost indefinitely, both in distribution (depth < 5) and out of distribution sequences. Below we paste the table version of Figure 7 (OOD accuracy only) for your quick reference. The sequential length ranges from 1 to 2560.
>
> | Model | 1-10 | 10-20 | 20-40 | 40-80 | 80-160 | 160-320 | 320-640 | 640-1280 | 1280-2560 |
> |-------|------|------|------|------|------|------|------|------|------|
> | LSTM | 100.0 ± 0.0 | 95.6 ± 0.7 | 53.6 ± 0.8 | 50.0 ± 0.0 | 50.0 ± 0.0 | 50.0 ± 0.0 | 50.0 ± 0.0 | 50.0 ± 0.0 | 50.0 ± 0.0 |
> | Mamba2 | 100.0 ± 0.0 | 100.0 ± 0.0 | 91.3 ± 6.8 | 61.2 ± 11.7 | 50.7 ± 0.7 | 50.1 ± 0.1 | 50.1 ± 0.1 | 50.1 ± 0.2 | 50.1 ± 0.1 |
> | Transformer | 100.0 ± 0.0 | 99.8 ± 0.1 | 86.6 ± 1.8 | 64.8 ± 1.8 | 51.0 ± 0.4 | 50.0 ± 0.0 | 50.0 ± 0.0 | 50.0 ± 0.0 | 50.0 ± 0.0 |
> | Ours | **100.0 ± 0.0** | **100.0 ± 0.0** | **100.0 ± 0.0** | **100.0 ± 0.0** | **99.9 ± 0.1** | **99.7 ± 0.5** | **99.0 ± 1.3** | **97.8 ± 2.9** | **95.9 ± 5.1** |
>
>
>
>
>
> ## “Why AKOrN-based framework? “
>
> There were several concerns regarding why we experimented on the model based on Kuramoto model.
> As we clarified above, the goal of our work is to extend the capabilities of recurrent models for the OOD generalization on the tasks with sequential streaming input.
> It was our hope that we can particularly extend the ability of the recent recurrent models to synchronize the internal loop with the static input (Geshkovski et al., 2023; Miyato et al., 2025), thereby **encouraging the model to consolidate the past inputs (context) to organized states that can help the model reason over a long horizon.**
>
> To test our intuition, we additionally examined the trajectory of energy and latent states, and confirmed that our models are in fact iteratively consolidating the sequential inputs so that it can promote the task performance.
> For example, for the Dyck task, the trajectories of the energies and states are in alignment with the change of the states in the stack of brackets (Figure 7 left), and they are forming a structured pattern without the need of specific prior knowledge of the task.
> More importantly, our model with such a naturally trained  “consolidation ability”  can generalize significantly better than other baselines (Figures 7 right, 8).
> We were able to observe similar patterns in the RL setting as well (Figure 10).
> Our observation warrants further investigation of this mechanism, because it may play an important role in solving challenging real-world applications with inherently streaming input characteristics, as commented by the reviewer KmRT.
>
> ## More baselines.
>
> We added four baselines — looped Transformer (Fan et al., 2025), S5 (Smith et al., 2022), and CTM (Darlow et al., 2025\) — to the maze task, and confirmed that our approach performs better in OOD generalization. To adapt the looped Transformer to the stream of sequential input, we simply applied it to the sequence that grows in length every time a new input arrives.
>
> | Method         | ID Accuracy (%)   | OOD Accuracy (%)   |
> |----------------|-------------------|--------------------|
> | LSTM           | 52.7 ± 9.6        | 3.3 ± 1.8          |
> | Mamba          | 20.9 ± 0.4        | 19.6 ± 1.9         |
> | TF             | 58.6 ± 4.2        | 10.6 ± 6.1         |
> | Looped TF      | 21.4 ± 0.1        | 19.9 ± 0.5         |
> | S5             | 7.6 ± 0.1         | 6.1 ± 0.3          |
> | CTM            | 11.2 ± 0.1        | 7.7 ± 0.4          |
> | Ours        | **96.8** ± 0.1        | **61.2** ± 1.9         |
>
> We hypothesize that our approach performs better on the sequential stream because every time we receive the new input, the model synchronizes it with the latents that have gone through the aforementioned recursive “consolidation” process at a faster time scale.
> We are different from the usual transformer type approach in that we carry over the result of consolidation to the next step; please see Figure 1.

---

> > ### Author Response · Authors · 2025-11-26
> > **Response to all reviewers Part3**
> >
> > ## Additional ablation on the memory mechanism
> >
> > In response to the concern regarding the choice of the memory type suited for the task of interest, we conducted an additional maze experiment to confirm that “using both types of memory” can cover both of the cases we mentioned in our work.  We added this ablation result in Figure 5\. Below we paste the table version of the figure.
> >
> > | Method         | ID Accuracy (%)   | OOD Accuracy (%)   |
> > |----------------|-------------------|--------------------|
> > | Ours(V)        | 65.9 ± 17.2       | 38.6 ± 3.4         |
> > | Ours(E)        | 93.9 ± 1.3        | 57.8 ± 2.0         |
> > | Ours(I)        | 96.8 ± 0.1        | 61.2 ± 1.9         |
> > | Ours(IE)       | 95.8 ± 1.5        | 57.9 ± 4.3         |
> >
> > `Ours(V)`, `Ours(I)`, `Ours(E)`, and `Ours(IE)` respectively denote our model equipped with no memory, internal memory, external memory, and both.
> > We plan to add more ablations with the remaining time allowing.

---

### Author Response · Authors · 2025-12-03
**New: Additional Results**

We thank the reviewers and AC again for their careful reading and for the many concrete suggestions. In this follow‑up, we report three additional results addressing the remaining concerns.

### **Additional ablation of memory modules on MiniGrid (LavaCrossing)**

Reviewers dSwD and kmRT asked how robust our conclusions are to the choice of internal vs. external memory. To address this, we ran the same memory ablation as in the maze experiment, but now on the MiniGrid task. Due to the resource limitations, we only compared the single-memory and the double-memory variants.

| Env | LSTM | Mamba | TransformerXL | Ours(E) | Ours(IE) |
|----|----|----|----|----|----|
| S9N1 (ID) | 57.1 ± 52.1 | 75.1 ± 42.0 | 47.0 ± 44.7 | 75.4 ± 42.2 | 92.9 ± 3.2 |
| S9N2 (OOD) | 49.6 ± 47.5 | 66.3 ± 39.9 | 44.6 ± 45.8 | 75.1 ± 42.0 | 85.2 ± 12.5 |
| S9N3 (OOD) | 29.6 ± 28.1 | 52.0 ± 37.5 | 40.2 ± 40.6 | 74.9 ± 41.9 | 76.4 ± 27.4 |
| S11N5 (OOD) | 12.4 ± 12.3 | 24.1 ± 17.0 | 21.3 ± 23.7 | 64.7 ± 40.2 | 50.8 ± 32.5 |

As in the maze task, the configuration with **both** memories `Ours(IE)` performs on par with the best single‑memory variant `Ours(E)`. This confirms that using both memory types is a robust default also in the RL setting and further resolves the concern that our method would require task‑specific hand‑tuning of the memory choice.

### Significance and robustness of MiniGrid results

Reviewer Cbwp expressed concerns about the small number of seeds in the MiniGrid tasks. We now **increased the number of seeds from 3 to 5 for all MiniGrid tasks** and **added training curves of success rates over environment steps (Figure 11)**. The new plots make it explicit that (i) our method remains competitive or superior to the baselines across tasks when averaged over more seeds, and (ii) the gains are not artifacts of a particular random seed but arise from consistently faster and/or higher convergence. We believe these additions strengthen the statistical robustness of our RL results and clarify the practical significance of the proposed architecture in partially observable tasks.

### New result: comparison with frontier LLMs on Dyck.

Reviewer 7zsu asked whether our synthetic benchmark is informative for current foundation models and how our architecture compares to current foundation models. In the revision we conducted a new experiment that evaluated three frontier LLMs—Gemini 3.0 Pro, Claude 4.5 Opus, and GPT‑5.1 Thinking—on the same Dyck‑(30,5) task. The LLMs are provided with the *ground‑truth stack‑based algorithm* in the prompt and are instructed to follow the deterministic procedure (App. C.1). The table below summarizes the results.

| Model                    | 1-10 | 10-20 | 20-40 | 40-80 | 80-160 | 160-320 | 320-640 | 640-1280 | 1280-2560 | 2560-5120 | 5120-10240 | 10240-20480 | 20480-40960 | 40960-81920 | 81920-163840 |
|--------------------------|------|-------|-------|-------|--------|---------|---------|----------|-----------|-----------|------------|-------------|-------------|-------------|--------------|
| Gemini 3.0 Pro  | 77.8 | 58.7  | 51.2  | 44.0  | 36.6   | 30.6    | 23.7    | N/A      | N/A       | N/A       | N/A        | N/A         | N/A         | N/A         | N/A          |
| Claude 4.5 Opus          | 96.7 | 92.3  | 90.0  | 78.9  | 73.5   | 57.2    | N/A     | N/A      | N/A       | N/A       | N/A        | N/A         | N/A         | N/A         | N/A          |
| GPT 5.1 Thinking         | 98.2 | 93.8  | 87.4  | 76.2  | 59.9   | 45.0    | 34.1    | N/A      | N/A       | N/A       | N/A        | N/A         | N/A         | N/A         | N/A          |
| Ours                     | **100.0** | **100.0** | **99.9** | **100.0** | **100.0** | **99.3** | **98.5** | **97.6** | **97.2** | **97.0** | **96.0** | **94.8** | **94.6** | **92.7** | **90.9** |

We see that all three LLMs perform well at the beginning of the stream but their accuracy deteriorates steadily as the position advances. In contrast, our model maintains ≥90% accuracy even in the longest position. The LLM predictions terminate once their token limit is reached (as indicated by "N/A"), whereas our model has no intrinsic context limit and can be run indefinitely. The result justifies our use of Dyck as a diagnostic benchmark and demonstrates that the capability we study, long‑horizon streaming reasoning under fixed‑size state, is not yet solved by current LLMs.

---

### Meta-Review · Area_Chair_cQPx · 2026-01-08

**Summary:**

This work presents an approach for incorporating real-time reasoning into recurrent neural networks. The authors have provided an extensive rebuttal addressing the comments from eight reviewers, and we appreciate their hard work. However, the reviewers still raise concerns regarding the clarity of the paper’s presentation. While the idea itself is interesting, the distinctions from traditional approaches are not sufficiently clarified, and the experimental setup requires further improvement.

Although the proposed approach is promising, the paper is not recommended for acceptance in its current form. The authors are encouraged to address the reviewers’ feedback and further strengthen the work for resubmission to other venues.

**Reviewer Concerns:**

The paper presentation needs to be improved.

**Reviewer Scores:**

NA

---

### Decision · Program_Chairs · 2026-01-26

Reject